# An Information-Theoretic Lower Bound on the Generalization Error of Autoencoders

**Shyam Venkatasubramanian**\*                    *shyam.venkatasubramanian@duke.edu*
*Department of Electrical and Computer Engineering*
*Duke University*

**Sean Moushegian**\*                    *sean.moushegian@duke.edu*
*Department of Electrical and Computer Engineering*
*Duke University*

**Ahmed Aloui**\*                    *ahmed.aloui@duke.edu*
*Department of Electrical and Computer Engineering*
*Duke University*

**Vahid Tarokh**                    *vahid.tarokh@duke.edu*
*Department of Electrical and Computer Engineering*
*Duke University*

**Reviewed on OpenReview:** `https://openreview.net/forum?id=0esF0M467w`

## Abstract

Quantifying the limitations of classical neural network architectures is a critically underexplored area of machine learning research. Deriving lower bounds on the optimal performance of these architectures can facilitate improved neural architecture search and overfitting detection. We present an information-theoretic lower bound on the generalization mean squared error of autoencoders with sigmoid activation functions. Through the Estimation Error and Differential Entropy (EEDE) inequality for continuous random vectors, we derive this lower bound, which provides a new perspective on the inherent limitations and capabilities of autoencoders. Our analysis extends to the examination of how this lower bound is influenced by various architectural features and data distribution characteristics. This study enriches our theoretical understanding of autoencoders and has substantial practical implications for their design, optimization, and application in the field of deep learning.

## 1 Introduction

Autoencoders, originating in the late 1980s (see Ballard (1987)), have undergone significant evolution alongside the development of deep neural networks (Sewak et al., 2020; Pidhorskyi et al., 2020; Georgescu et al., 2023; Esser et al., 2023). Given a random variable, $X$, supported on $\mathcal{X} \subseteq \mathbb{R}^d$, we consider a training dataset that is sampled from $X \sim D$, where $\mathcal{I} = \{x^{(i)}\}_{i=1}^N$. An autoencoder aims to compress a sample, $x \in \mathbb{R}^d$, into a low-dimensional latent representation, $y \in \mathbb{R}^l$, and then form a reconstructed sample, $\hat{x} \in \mathbb{R}^d$, with the goal of making $\hat{x}$ as similar as possible to $x$. Although an autoencoder is trained to learn the identity mapping $\zeta(x) = x$, the bottleneck imposed by its low-dimensional latent layer forces it to learn, instead, a function $\hat{\zeta}$ that approximates $\zeta$ in expectation over the data distribution, $\mathcal{D}$. This encourages the autoencoder to learn the features of samples, $x$, that best explain the information in $\mathcal{I}$. Accordingly, autoencoders can be viewed as a method of nonlinear, trainable dimensionality reduction.

---

\*Equal Contribution. Github Repository: `https://github.com/shyamven/InformationTheoreticLowerBounds`

The concept of learning low-dimensional representations is closely related to compression, and has spurred several studies at the intersection of information theory and deep autoencoders (Baldi, 2012; Yu & Principe, 2019). Correspondingly, there have been several investigations into the relationships between information theory and generalized neural networks (Tishby & Zaslavsky, 2015; Neelakanta, 2020; Shwartz Ziv & LeCun, 2024). We focus on exploring the limits of the approximation capabilities of autoencoders.

One of the most fundamental questions concerning multi-layer perceptrons (MLPs) pertains to the extent of their capacity to accurately approximate the true target function. Cybenko (1989) proved the Universal Approximation Theorem, which establishes that MLP networks are *expressive*. Specifically, any MLP network, $\psi$, that consists of one hidden layer with $n$ perceptrons, can arbitrarily approximate any continuous target function, $\xi$, as $n$ tends to infinity. Due to practical computational constraints, however, any implementable MLP will have finite $n$.

Recently, there has been a growing interest in deriving generalization bounds on the approximation capabilities of neural networks. In this context, we define the *generalization error* as the population risk, which is the expected loss over the data distribution. This is distinct from the term *generalization gap* (the difference between the empirical risk and the population risk) (Jiang et al., 2019). Various generalization upper bounds have been proposed in Vapnik (1968), Valiant (1984), and Mustafa et al. (2024). However, these results often provide worst-case, probabilistic guarantees. In contrast, lower bounds on the generalization error reveal the fundamental limits of what is achievable, which only few works have addressed (Seroussi & Zeitouni, 2022). We aim to leverage information theory to derive a lower bound on the generalization error of autoencoders.

In this study, we prove a result for autoencoders that is analogous to an inverse of Cybenko's theorem: we demonstrate that for a large class of autoencoders with a finite number of perceptrons per layer, $n$, equality between $\zeta$ and the autoencoder, $\mathcal{A}$, is not achievable. More specifically, we establish a lower bound on the generalization error that is a function of the architecture of $\mathcal{A}$ and the differential entropy of $\mathcal{D}$. This is an objective condition that cannot be violated even with infinite data and perfect optimization and training.

**Motivation.** The conceptualization of this lower bound would lend to several valuable insights which are of interest to the deep learning community at large. In particular, finding the smallest architecture that can attain some threshold level of performance at a task is a common objective in neural architecture search, but creating numerous candidate architectures of various sizes, and training each of them to measure performance, is computationally expensive. By establishing a lower bound on the generalization error, one can shrink the space of architectures to be explored *before any training* by guaranteeing that some architectures cannot achieve a given performance threshold. Furthermore, overfitting is a serious problem in deep learning, and many characterizations of overfitting are empirical. A lower bound on the generalization error would provide an objective condition to detect overfitting in autoencoders with *theoretical guarantees*: whenever the training loss descends below this bound, the fit is *theoretically* too good to be true and will not generalize. We further hypothesize that this bound can be leveraged to suggest overfitting in regression and classification tasks. We present empirical results for both neural architecture search and overfitting detection in Section 7.

## 2 Background

**Autoencoders.** An autoencoder, $\mathcal{A}_\theta$, is an MLP comprising input and output layers each of dimension $d$, parameterized by $\theta$. All autoencoders have at least one hidden layer, and may have many more; we refer to the hidden layer with the smallest dimensionality, $l$, as the latent layer*. We express the autoencoder using the composition $\mathcal{A}_\theta(\cdot) = g(f(\cdot))$, where $f : \mathbb{R}^d \mapsto \mathbb{R}^l$ and $g : \mathbb{R}^l \mapsto \mathbb{R}^d$. Autoencoders are commonly trained with the empirical MSE loss, depicted in Eq. (1):

$$\mathcal{L}(\mathcal{A}_\theta, \mathcal{I}) = \frac{1}{|\mathcal{I}|} \sum_{x \in \mathcal{I}} \|x - \mathcal{A}_\theta(x)\|_2^2. \tag{1}$$

---

*We note that this lower bound is valid for any choice of hidden layer. We restrict our analysis to the latent layer as it yields the most informative bound.

Optionally, regularization techniques can be added to the training loss objective. We note that although the empirical MSE is the most common training loss objective, our lower bound is on the generalization MSE and is independent of the training methodology.

In practice, $\mathcal{A}_\theta$ is leveraged for dimensionality reduction, mapping inputs in a high $d$-dimensional space to some low $l$-dimensional representation. It attempts to preserve enough information about the inputs in this low dimensional representation for reconstruction.

**Information-Theoretic Inequalities.** Consider a Markov chain $X \mapsto Y \mapsto \hat{X}$, such that $X, \hat{X} \in \mathcal{X}$, and $Y \in \mathcal{Y}$. We recall information-theoretic bounds on the reconstruction $\hat{X}$ of $X$. Fano's Inequality, where $\mathcal{X}, \mathcal{Y}$ are discrete finite supports of discrete, scalar random variables, $X, Y$, states that:

**Theorem 2.1** (Fano's Inequality). *(Cover & Thomas, 2006, Theorem 2.10.1, p. 38)*

$$\mathbb{P}[\hat{X} \neq X] \geq \frac{H(X|Y) - 1}{\log |\mathcal{X}|}, \tag{2}$$

where $|\mathcal{X}|$ is the cardinality of the support $\mathcal{X}$ of $X$, and $H(X|Y)$ denotes the conditional discrete entropy in bits. The continuous analog of Fano's inequality is the Estimation Error and Differential Entropy (EEDE) inequality, which for uncountable supports, $\mathcal{X}, \mathcal{Y}$, of continuous, scalar random variables, $X, Y$, states that:

**Theorem 2.2** (EEDE Inequality). *(Cover & Thomas, 2006, Theorem 8.6.6. p. 255)*

$$\mathbb{E}\left[(\hat{X} - X)^2\right] \geq \mathrm{Var}(X|Y) \geq \frac{1}{2\pi e} e^{2h(X|Y)}, \tag{3}$$

where $h(X|Y)$ denotes the conditional differential entropy in nats.

**Form of Bound.** Our goal is to derive an information-theoretic lower bound on the generalization MSE of an autoencoder as a function of its architecture and the complexity of the data distribution, $\mathcal{D}$, utilizing the inequalities described in Section 2. We propose that this lower bound is of the form:

$$\mathbb{E}\left[\|X - g(f(X))\|^2\right] \geq \mathcal{F}(d, l, K, h_\mathcal{D}), \tag{4}$$

where $d$ is the dimension of the input space, $l$ is the dimension of the latent space, and $h_\mathcal{D}$ is the differential entropy of the data distribution, $\mathcal{D}$, which may be analytically calculated or empirically estimated (we further explore this in Section 5.2). Moreover, $K = \sup_\theta K_\theta$, where $K_\theta$ denotes the Lipschitz constant of the decoder with respect to the latent input, for autoencoder parameterization, $\theta$.

## 3 Lower Bound From Injective Noise

In this section, we derive a lower bound on the generalization MSE of $\mathcal{A}_\theta$. For simplicity, we restrict ourselves to the case where $\mathcal{D}$ is supported on $\mathcal{X} = [0, 1]^d$, which is common for image datasets with bounded pixel intensities. We further restrict ourselves to autoencoders using sigmoid activation functions after each layer.

We first suppose that an autoencoder characterizes a [deterministic] Markov chain, $X \mapsto Y \mapsto \hat{X}$, wherein $X, \hat{X} \in \mathcal{X}, Y \in \mathcal{Y} = [0, 1]^l$. To formalize our lower bound, we revisit Section 2, and choose an appropriate inequality. In machine learning literature, many datasets are recorded as observations of a discrete random variable (e.g., for image datasets, a pixel takes one of 256 discrete intensity levels). While Fano's Inequality of Theorem 2.1 might seem suitable, the enormity of $|\mathcal{X}|$ for high-dimensional datasets precludes any useful bound. In our approach, we circumvent this issue by viewing the samples as existing in a continuous space, recorded as discrete observations only for numerical convenience, wherein we can apply the EEDE inequality of Theorem 2.2. Consequently, the primary challenge becomes the precise formulation of $h(X|Y)$ in these high-dimensional domains, as the estimation of $\mathrm{Var}(X|Y)$ in such settings is intractable.

Even this approach encounters a fundamental obstacle, as the encoder is deterministic. One might consider the identity that $h(X|Y) = h(X) - h(Y) + h(Y|X)$. However, since $Y = f(X)$, it follows that $h(Y|X) = -\infty$.

Thus, the EEDE inequality yields a trivial and non-informative lower bound, stating that the generalization MSE is bounded from below by zero.

To circumvent this issue and determine a meaningful lower bound, we propose a novel approach involving the introduction of a small injective noise, $Z \in \mathbb{R}^l$, into the latent representation before decoding *solely as a proof technique.* By doing so, we transform the autoencoder into a modified "noisy" version that incorporates some stochasticity, which enables us to bypass the limitations imposed by the deterministic nature of $f(X)$. We then account for the effects of this small noise, $Z$, to arrive at a lower bound on the generalization MSE of the *original, unperturbed "noiseless" autoencoder.* The complete derivation of the lower bound leveraging this modified autoencoder framework is provided in Section A of the Appendix. We now obtain an expression for the MSE of the noiseless autoencoder by expanding the squared-L2 norm.

$$\underbrace{\mathbb{E}\left[\|g(f(X)) - X\|^2\right]}_{\text{Term 1}} = \underbrace{\mathbb{E}\left[\|g(f(X)) - g(f(X) + Z)\|^2\right]}_{\text{Term 2}} + \underbrace{\mathbb{E}\left[\|X - g(f(X) + Z)\|^2\right]}_{\text{Term 3}}$$
$$- \underbrace{\mathbb{E}\left[2[g(f(X)) - g(f(X) + Z)]^T[X - g(f(X) + Z)]\right]}_{\text{Term 4}} \tag{5}$$

From Eq. (5), random variable $Z$ is our injective noise with variance $\sigma^2$: $Z \sim \mathcal{N}(0_l, \sigma^2 I_l)$, where $0_l \in \mathbb{R}^l$ is a vector of zeros and $I_l$ is the $l \times l$ identity matrix. Term 1 is the MSE of the original noiseless autoencoder. Term 2 captures the difference in reconstruction between the noisy and noiseless decoders. Term 3 denotes the MSE of the noise-injected autoencoder. Term 4 is a cross-term.

**Remark 3.1.** *To obtain a lower bound on the MSE of the noiseless autoencoder, we form an upper bound on the cross-term, and lower bounds on the MSE of the noise-injected autoencoder and between the noisy and noiseless decoders.*

We bound Terms 2, 3, and 4 in the following Lemmas, to obtain the overall lower bound in Section 3.1.

**Remark 3.2.** *We have that:*

$$\mathbb{E}\left[\|g(f(X)) - g(f(X) + Z)\|^2\right] \geq 0. \tag{6}$$

*Proof.* Proof is detailed in Section A.1 of the Appendix. □

**Lemma 3.3.** *We have that:*

$$\mathbb{E}\left[\|X - g(f(X) + Z)\|^2\right] \geq \frac{d}{2\pi e} \exp\left(\frac{2}{d}\left[h_{\mathcal{D}} - \frac{l}{2}\log\left(2\pi e\left(\frac{1}{4} + \sigma^2\right)\right) + \frac{l}{2}\log(2\pi e\sigma^2)\right]\right), \tag{7}$$

*where $h_{\mathcal{D}}$ is the differential entropy of distribution $\mathcal{D}$.*

*Proof.* Proof is detailed in Section A.2 of the Appendix. □

**Lemma 3.4.** *We have that:*

$$\mathbb{E}\left[2[g(f(X)) - g(f(X) + Z)]^T[X - g(f(X) + Z)]\right] \leq 2\sqrt{(K^2 l\sigma^2)(d)} = 2K\sqrt{ld\sigma^2},$$

*where $K$ is the upper bound on the Lipschitz constant of $g$.*

*Proof.* Proof is detailed in Section A.3 of the Appendix. □

### 3.1 Lower Bound on Noiseless Autoencoder

We have now bounded each of the terms comprising Eq. (5). Accordingly, to obtain a lower bound on the generalization MSE of the original noiseless autoencoder, we substitute these bounds for the terms in Eq. (5).

Hereafter, for brevity, we define:

$$s \triangleq \sigma^2, \qquad \beta \triangleq \exp\left(\frac{2h_{\mathcal{D}}}{d}\right), \qquad \alpha \triangleq \frac{d}{2\pi e}, \qquad \gamma \triangleq 2K\sqrt{ld}, \tag{8}$$

where $\alpha, \beta, \gamma$ do not depend on $s$.

**Theorem 3.5.** *The lower bound on the generalization MSE of the noiseless autoencoder is given by:*

$$\mathcal{F}(s, d, l, K, h_{\mathcal{D}}) = \alpha\beta\left(\frac{s}{\frac{1}{4} + s}\right)^{\frac{l}{d}} - \gamma\sqrt{s}. \tag{9}$$

*Proof.* Proof is detailed in Section A.4 of the Appendix. $\square$

We observe our lower bound function is defined $\forall s \in [0, \infty)$, and that the inequality of Theorem 3.5 holds for any nonnegative $s$. We now aim to find the value of $s$ which maximizes $\mathcal{F}$ and we denote this optimal noise power level as $s^*$. Since the derivative of $\mathcal{F}$ with respect to $s$ appears to be mathematically intractable, we instead study a numerical approximation, $\hat{s}^*$, of $s^*$.

## 4 Properties of the Lower Bound

In this section, we study the shape of the bound function, $\mathcal{F}$, with respect to $s$. Under "practical" architectures (which we will define more formally), we see that when $l$ is large, the bound function, $\mathcal{F}$, is non-positive for all $s$ (and thus non-informative), and when $l$ is small, the bound function is positive for some $s$, ruling out the possibility of perfect reconstruction. This mathematical relationship captures the notion that when the latent layer is of larger dimension, more information about the original sample, $x$, can be retained, thus yielding improved reconstruction. Finally, we provide an approximation, $\hat{s}^*$, of $s^*$, which maximizes $\mathcal{F}$.

We begin by defining the maximum value achieved by the MSE lower bound function, $\mathcal{F}_\eta^*$, as follows:

$$\mathcal{F}_\eta^* = \max_{s \in [0, \infty)} \mathcal{F}_\eta(s), \qquad \text{where:} \qquad \eta = (d, l, K, h_{\mathcal{D}}). \tag{10}$$

For analytical purposes, suppose that $d, l \in \mathbb{Z}^+, K \in \mathbb{R}^+, h_{\mathcal{D}} \leq 0$, and $s \geq 0$. Analyzing the extremes of $\mathcal{F}_\eta$:

**Lemma 4.1.**

$$\mathcal{F}_\eta(0) = 0, \quad \lim_{s \to \infty} \mathcal{F}_\eta(s) = -\infty, \quad and: \quad \mathcal{F}_\eta(s) \leq d. \tag{11}$$

*Proof.* Proof is detailed in Section B.1 of the Appendix. $\square$

Together with the fact that $\mathcal{F}_\eta$ is continuous in $s$, these notions guarantee that $\mathcal{F}_\eta$ is bounded from above, and that $\mathcal{F}_\eta^*$ exists and is finite. We note the bound, $\mathcal{F}_\eta \leq d$, aligns with $\mathbb{E}[\|\hat{X} - X\|^2] \leq d$. Next, we observe:

**Lemma 4.2.** *For fixed $s \geq 0$:*

$$\frac{\partial \mathcal{F}_\eta(s)}{\partial K} \leq 0, \quad and: \quad \frac{\partial \mathcal{F}_\eta(s)}{\partial h_{\mathcal{D}}} \geq 0, \tag{12}$$

*where $\mathcal{F}_\eta(s)$ is a decreasing function of $l$.*

*Proof.* Proof is detailed in Section B.2 of the Appendix. $\square$

These properties verify the behavior we expect to observe in autoencoders. We see that as the expressivity of the decoder increases, the lower bound decreases, as the differential entropy of the distribution increases, the lower bound increases, and as the latent dimensionality decreases, the lower bound increases. Specifically, for non-extreme, "practical" $h_{\mathcal{D}}$ and $K$, there is a direct relationship between $l, d$ and the positiveness of $\mathcal{F}_\eta^*$.

**The *Practical* Regime.** If the distribution over the input samples is degenerate or of countable support ($h_{\mathcal{D}} = -\infty$), then even when $l = 1$, an injective encoder function, $f$, can be learned, since $Y$ exists on an uncountable support for any $l \geq 1$, whereby there cannot be a positive lower bound. Moreover, if the upper bound on the Lipschitz constant, $K$, is constrained to be very close to zero, the decoder lacks expressivity, yielding overly compressed latent representations and large reconstruction errors. We consider these specific scenarios (the "impractical" regime) in Section B.4 of the Appendix for completeness, where we argue that these cases are of no interest. Subsequently, we consider the more common practical regime, which we define as the regime where Assumptions 4.3 and 4.4 hold simultaneously.

**Assumption 4.3.** $h_{\mathcal{D}} > -\infty$

**Assumption 4.4.** $K \geq (\pi e \sqrt{2})^{-1} \approx 0.083$

We now present a result in the practical regime:

**Theorem 4.5.** *We have that under Assumption 4.3 and Assumption 4.4:*

$$l < \frac{d}{2} \iff \mathcal{F}_{\eta}^* > 0. \tag{13}$$

*Proof.* Proof is detailed in Section B.3 of the Appendix. □

This result demonstrates that under reasonable assumptions, wherein the decoder, $g$, is sufficiently expressive, and the data distribution, $\mathcal{D}$, has finite differential entropy, the most informative generalization error bound posited by our lower bound function is strictly positive when the latent dimensionality is less than half the input dimensionality. We illustrate the positiveness of our lower bound in Figure 1.

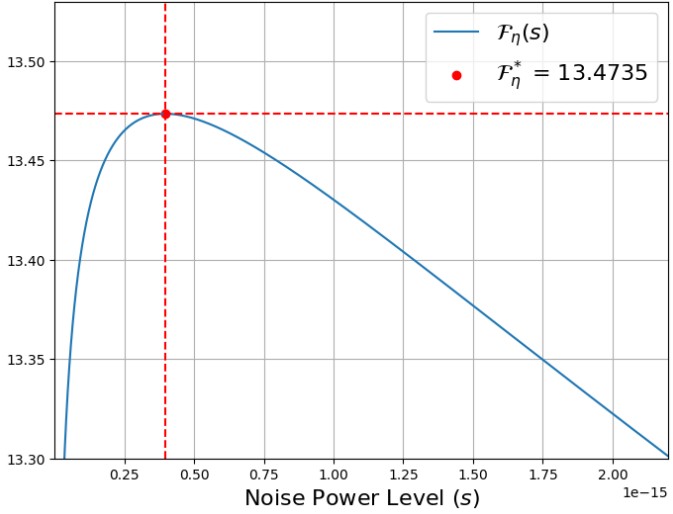

Figure 1: MSE lower bound as a function of $s$ for $d = 784, l = 10, h_{\mathcal{D}} = -300$, and $K = 10^5$.

**Most Informative Bound.** In the practical regime, the decoder (with moderate to high $K$) is sufficiently expressive, wherein even a small amount of injective noise within the latent layer substantially perturbs the autoencoder reconstruction. Accordingly, the quantity $g(f(X)) - g(f(X) + Z)$ from the cross-term of Eq. (5) becomes extremely large as $s$ increases, especially for large $K$. This implies that $s^*$ must be very small, many orders of magnitude less than one. Under the relatively mild assumption that $s^* << \frac{1}{4}$, we can leverage the following approximation:

$$\frac{s}{s + \frac{1}{4}} \sim 4s, \quad \text{as} \quad s \to 0^+. \tag{14}$$

Utilizing this approximation, we note that the following result holds, under which we obtain a valid approximation, $\hat{\mathcal{F}}_{\eta}^*$, of the most informative bound, $\mathcal{F}_{\eta}^*$.

**Theorem 4.6.** *The value $s^*$ of $s$ which maximizes $\mathcal{F}_\eta$ can be approximated by $\hat{s}^*$, wherein $\mathcal{F}_\eta^*$ from Eq. (10) can be approximated by $\hat{\mathcal{F}}_\eta^*$, such that:*

$$\hat{s}^* = \left(\frac{4^d \gamma d}{2 \cdot 4^l \alpha \beta l}\right)^{\frac{2d}{2l-d}}, \qquad and: \qquad \hat{\mathcal{F}}_\eta^* = \alpha\beta\left(4\hat{s}^*\right)^{\frac{l}{d}} - \gamma\left(\hat{s}^*\right)^{-\frac{1}{2}}. \tag{15}$$

*Proof.* Proof is detailed in Section B.5 of the Appendix. $\qquad\square$

**Remark 4.7.** *Since $\hat{\mathcal{F}}_\eta^*$ is a conservative estimate of $\mathcal{F}_\eta^*$, it follows that $\hat{\mathcal{F}}_\eta^*$ is a valid lower bound on the generalization MSE of $\mathcal{A}_\theta$.*

# 5 Estimation of Bound Parameters

While $d, l$ are known, the bound in Theorem 3.5 is also a function of $K$ and $h_\mathcal{D}$. In this section, we propose methods of estimating these quantities for $l < d/2$.

## 5.1 Bounding the Lipschitz Constant

Since $g$ is an MLP comprising sigmoid activation functions, we know that $g$ is Lipschitz continuous for some $K_\theta$. We recall that since $K_\theta$ is parameterized by $\theta$, it is a function of the trained $g$. As we make no claims regarding the parameterization of $g$ or its training, we must derive a $K$ such that any learnable $g$ of a fixed architecture is $K$-Lipschitz. As such, rather than form a point estimate of $K_\theta$, we estimate an upper bound, $K \colon K_\theta \leq K$, for a fixed architecture. By Lemma 4.2, when we overestimate $K_\theta$, we underestimate $\mathcal{F}_\eta$, which guarantees the reliability of the bound.

We now study a particular parametrization, $\theta$, of $\mathcal{A}_\theta$, by considering $K_\theta$ in place of $K$ from Theorem 3.5 to demonstrate that the MSE lower bound is insensitive to overestimates of $K_\theta$. When $K_\theta$ is large, we observe that the optimal injective noise power level, $s^*$, is sensitive to changes in $K_\theta$. The maximum value achieved by the MSE lower bound, however, is not, wherein changes that span several orders of magnitude when $K_\theta$ is large yield a similar $\mathcal{F}_\eta^*$.

**Theorem 5.1.** *For the optimal injective noise power level, $s^*$, $\left|\frac{\partial \mathcal{F}_\eta^*}{\partial K_\theta}\right| \leq 2\sqrt{lds^*}$ and $\lim_{K_\theta \to \infty} s^* = 0$.*

*Proof.* Proof is detailed in Section C.1 of the Appendix. $\qquad\square$

Suppose the decoder $g$ has $L-1$ hidden layers. We assume that $\forall k : 1 \leq k \leq L$, the dimensionality of layer $k$ is $c_k \leq d$, which is a standard constraint for decoders. Each layer, $k$, is parameterized by $\theta_k \in \mathcal{M}_{c_k \times c_{k+1}}(\mathbb{R})$. We recall maximum norm regularization, which imposes that $\|\theta_k\|_F \leq M$, $\forall k$, and is a common technique utilized in deep learning to prevent overfitting during training (Srebro & Shraibman, 2005; Lee et al., 2010; Hinton et al., 2012; Goodfellow et al., 2013). Accordingly, we observe that:

**Theorem 5.2.** *Under maximum norm regularization,*

$$K_\theta \leq \frac{M^{2L}}{16^L} = K. \tag{16}$$

*Proof.* Proof is detailed in Section C.2 of the Appendix. $\qquad\square$

We observe that upper bounding the norm, $\|\theta_k\|_F$, is sufficient to establish an upper bound on $K_\theta$. Similar derivations have been presented in Béthune et al. (2024).

Without relying on maximum norm constraints, we can still bound $K_\theta$, drawing inspiration from the diminishing sensitivity of the sigmoid function for large inputs, due to the vanishing gradients phenomenon (Hochreiter, 1998; Roodschild et al., 2020). This effect stems from the computational limitations of machines, particularly with 64-bit floating-point representations, which struggle to accurately process exceedingly small values. These computational constraints further influence the maximal Lipschitz constant achievable.

**Theorem 5.3.** *In a decimal-64 floating point system,*

$$K_\theta \leq \sqrt{\left(\frac{10^{36}\, d^2}{16}\right)^L} = K.$$ (17)

*Proof.* Proof is detailed in Section C.2 of the Appendix. □

## 5.2 Estimating the Differential Entropy

The bound in Theorem 3.5 requires an estimate of the differential entropy, $h_\mathcal{D}$. For some distributions, $\mathcal{D}$, we can analytically calculate the differential entropy. In deep learning applications, however, we often encounter high-dimensional distributions for which this calculation is intractable, and worse still, we often do not have access to the density, $p(\cdot)$, of $\mathcal{D}$. Most commonly, we only have the dataset, $\mathcal{I}$ (Davis et al., 2011). Consider a sample, $x = (x_1, x_2, ..., x_d)$, with marginal variances $\sigma_i^2 = \text{Var}[X_i]$. We follow the general approach of Pichler et al. (2022) and construct a multivariate kernel density estimate of $p(x)$. To find reasonable parameters for our kernel density estimator, we populate the elements of a diagonal matrix, $\Sigma \in \mathcal{M}_{d \times d}(\mathbb{R})$, by leveraging Silverman's Rule of Thumb (Silverman, 1998):

$$h = \left[\left(\frac{4}{d+2}\right)^{\frac{1}{d+4}}\left(N^{\frac{-1}{d+4}}\right)\right], \quad \text{and:} \quad \Sigma = \begin{cases} h^2 \sigma_i^2, & i = j \\ 0 & i \neq j \end{cases}$$ (18)

Our kernel, $\mathcal{K}$, has the density of a Gaussian distribution with mean, $0_d \in \mathbb{R}^d$, and covariance, $\Sigma$, such that $\mathcal{K}(x) = \mathcal{N}(0_d, \Sigma)$. We now approximate $p(x)$ as:

$$p(x) \approx p_{\text{KDE}}(x) = \frac{1}{N}\sum_{i=1}^{N}\mathcal{K}(x - x^{(i)}).$$ (19)

Subsequently, we derive the estimate, $\hat{h}_\mathcal{D}$, of $h_\mathcal{D} \overset{\Delta}{=} -\mathbb{E}_{x \sim \mathcal{D}}[\log p(x)]$ via a sample mean estimator (in nats):

$$\hat{h}_\mathcal{D} = -\frac{1}{|\mathcal{I}|}\sum_{x \in \mathcal{I}}\log p_{\text{KDE}}(x).$$ (20)

## 6 Manifold Generalizations

In this section, we generalize our results to cases where the previously-state assumptions no longer hold. We introduce the notation $_K\mathcal{A}_{d \mapsto l \mapsto d}$ to denote an autoencoder with input dimension and output dimension, $d$, latent dimension $l$, and decoder Lipschitz constant upper bound, $K$.

### 6.1 Manifolds of dimension $m < d$

Many data distributions exist in a $m$-dimensional manifold of a $d$-dimensional ambient space, wherein $m < d$ (Ansuini et al., 2019). In this case, the differential entropy of $X = (X_1, X_2, \ldots, X_d)$ is $-\infty$. We now consider cases where $h_\mathcal{D} = -\infty$, but the differential entropy over the manifold is finite.

**Assumption 6.1.** *For data distribution $\mathcal{D}$ supported on $\mathcal{X} = [0,1]^d$, there exists disjoint sets $U, W$, where $|U| = m$, such that $U \cup W = \{1, 2, ..., d\}$, for which:*

$$\forall w \in W, \quad \exists \Lambda_w \quad s.t. \quad x_w = \Lambda_w(x_U),$$ (21)

*and:*

$$\forall u \in U, \quad h(X_u|(X_i)_{i \in U \setminus u}) > -\infty,$$ (22)

*where $\Lambda_w$, for $w \in W$, is a deterministic function of its arguments. Note that $U$ need not be unique.*

**Lemma 6.2.** *Under Assumption 6.1, $h(X_U) > -\infty$.*

*Proof.* Proof is detailed in Section D of the Appendix. □

Subsequently, we bound the reconstruction MSE of $_K\mathcal{A}_{d\mapsto l\mapsto d}$, and utilize the MSE lower bound of a reduced autoencoder, $_K\mathcal{A}_{m\mapsto l\mapsto m}$, in Theorem 6.3.

**Theorem 6.3.** *Suppose $\mathcal{D}$ violates Assumption 4.3 but obeys Assumption 6.1, and let $\mathbb{U}$ be the collection of all sets $U$ that satisfy Assumption 6.1. Consider the autoencoder $_K\mathcal{A}_{d\mapsto l\mapsto d}$. Then:*

$$\mathbb{E}[\|\ _K\mathcal{A}_{d\mapsto l\mapsto d}(X)-X\|^2] \geq \max_{U\in\mathbb{U}} \mathcal{F}^*_{(|U|,l,K,h(x_U))}. \tag{23}$$

*Proof.* Proof is detailed in Section D of the Appendix. □

## 6.2 Distributions with a Degenerate Marginal Component

In Assumption 4.3, we consider data distributions with finite differential entropy, but this is not realistic for many real-world distributions. For continuous distributions whose observations are recorded using binning, continuous marginal components with variances smaller than the bin resolution are clustered into common bins across the entire dataset, yielding effectively constant features. For example, in MNIST, the top-left pixel of each sample in $\mathcal{I}$ is the same (Deng, 2012). Thus, the differential entropy of MNIST will be $h_\mathcal{D} = -\infty$, violating Assumption 4.3 with $\mathcal{F}^*_\eta = 0$. This renders the estimation technique from Section 5.2 inapplicable.

We now propose a method to circumvent this issue to use the lower bound of Theorem 3.5. We can separate the set of marginal components into a set of zero-variance components, $W$, with the remaining components denoted $U$. While our bound is incalculable for $h_\mathcal{D} = -\infty$, we can bound the performance of an autoencoder using the finite differential entropy, $h(X_U) > -\infty$, where $X_U = (X_i)_{i\in U}$, as summarized in Corollary 6.4.

**Corollary 6.4.** *Let $(X_1, X_2, \ldots, X_d) = X \sim \mathcal{D}$, with $\sigma_i^2 = Var[X_i]$. Define $W \overset{\Delta}{=} \{i : 1 \leq i \leq d,\ s.t.\ \sigma_i^2 = 0\}$ and let $U \overset{\Delta}{=} \{1, 2, \ldots, d\}\backslash W$. Then:*

$$\mathbb{E}[\|\ _K\mathcal{A}_{d\mapsto l\mapsto d}(X) - X\|^2] \geq \mathcal{F}^*_{(|U|,l,K,h(X_U))}. \tag{24}$$

*Proof.* Proof is detailed in Section D of the Appendix. □

Often in practice, marginal components have near-zero variances that are several orders of magnitude smaller than the variances of the other features, which can lead to numerical instability when estimating $h_\mathcal{D}$. In this case, we permit an approximation and allow $W$ to be the set of components with exceedingly small variance below some set threshold, $\lambda \approx 0$, and apply Corollary 6.4. This approximation works well when $(X_i)_{i\in W}$ is independent of $X_U$.

# 7 Empirical Results

We now provide empirical results to illustrate several practical use cases of our lower bound. We first present how our lower bound can be leveraged for overfitting detection in reconstruction, regression, and classification tasks, and we also provide an application of our bound to the neural architecture search problem. We detail the datasets and neural network architectures utilized for each of these tasks in Section E of the Appendix.

## 7.1 Overfitting Detection

In numerous real-world tasks, data scarcity leads to small datasets that induce overfitting. In the following section, we employ our bound to detect overfitting.

### 7.1.1 Reconstruction Tasks

We begin by investigating the efficacy of the proposed lower bound for detecting overfitting in autoencoders trained on datasets with support $\mathcal{X} = [0, 1]^d$. This study utilizes the MNIST benchmark, where $x \in \mathcal{X}$, with

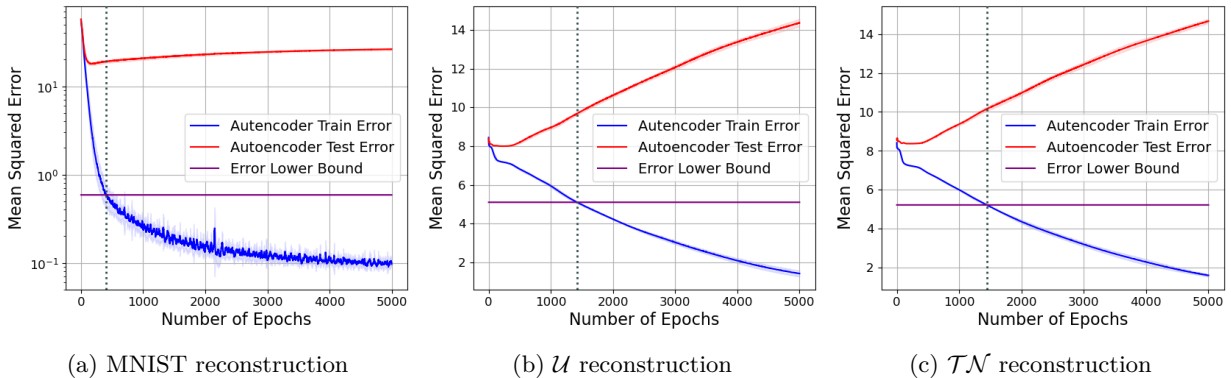

(a) MNIST reconstruction      (b) $\mathcal{U}$ reconstruction      (c) $\mathcal{TN}$ reconstruction

Figure 2: Overfitting detection results on MNIST, uniform distribution, and truncated normal distribution. The x-axis denotes the number of elapsed training epochs, while the y-axis indicates the mean squared error.

$d = 784$, and $l = 60$ (Deng, 2012). We estimate $h_{\mathcal{D}} \approx -494.5$ using the method of Section 5.2 for $\lambda = 10^{-2}$, due to the numerical issues with low-variance pixels in MNIST. We further study synthetic examples derived from a truncated normal ($\mathcal{TN}$) distribution, where $x_i \overset{\text{iid}}{\sim} \mathcal{TN}(0, 1, 0, 1)$, $\forall i \in \{1, \ldots, d\}$, with $d = 100$, $l = 2$, and $h_{\mathcal{D}} \approx -1.036$, and from a uniform distribution, $x \sim \mathcal{U}([0, 1]^d)$, with $d = 100$, $l = 2$, and $h_{\mathcal{D}} = 0$. For the synthetic distributions, $h_{\mathcal{D}}$ is calculated analytically. We now train an autoencoder on each of these datasets, employing $|\mathcal{I}| = 500$ training samples for MNIST, $|\mathcal{I}| = 50$ training samples for the synthetic datasets, and $|\mathcal{I}_{\text{val}}| = 10,000$ test samples. We impose a maximum norm constraint of $M = 2.25$ on the decoder's weights, so the decoder's Lipschitz constant has upper bound $K_\theta : K_\theta \le K \approx 0.1$ for $L = 2$ (per Theorem 5.2). We estimate the most informative bound, $\hat{\mathcal{F}}_\eta^*$ via Theorem 4.6, where $\eta = (d, l, K, h_{\mathcal{D}})$. Across all three datasets, we detect overfitting when the training MSE crosses $\hat{\mathcal{F}}_\eta^*$. At this point, the increasing test MSE corroborates the detection (see Figure 2). This result supports the practicality of the lower bound in detecting overfitting.

### 7.1.2 Regression and Classification Tasks

While our lower bound is directly applicable to overfitting detection on reconstruction tasks, we propose the following extension to regression and classification tasks. We recall that one objective of reconstruction is to compress the input data, $X$, while minimizing information loss within the latent space, $Y$. Analogously, in regression and classification tasks, we recall that the objective is to retain all relevant information from the input data, $X$, pertinent to the labels, $V$, in the latent space, $Y$ (Shwartz-Ziv & Tishby, 2017). We propose that for all such networks, $t(f(\cdot))$, where $t : \mathbb{R}^l \to \mathbb{R}^k$ and $\hat{V} = t(Y)$ denotes the predictions, there must exist a corresponding autoencoder, $\mathcal{A}_\theta(\cdot) = g(f(\cdot))$, formed by appending a decoder, $g$, to the original encoder, $f$. We hypothesize that the presence of overfitting in this corresponding autoencoder implies overfitting in $t(f(\cdot))$. To evaluate this claim, we reconsider the MNIST dataset with $|\mathcal{I}| = 20$ training samples and $k = 10$, and the synthetic $\mathcal{U}$ and $\mathcal{TN}$ datasets with $|\mathcal{I}| = 50$ training samples and $k = 1$, where $K \approx 0.1$. Per Figure 3, we empirically observe that overfitting in the autoencoder, $\mathcal{A}_\theta$, accompanies overfitting in $t(f(\cdot))$.

## 7.2 Neural Architecture Search

Regarding the context of neural architecture search (NAS) for autoencoders, selecting a lightweight architecture with generalization MSE that satisfies a predefined threshold, $\tau > 0$, is essential for ensuring effective data compression. For fixed input dimension, $d$, differential entropy, $h_{\mathcal{D}}$, and decoder Lipschitz upper bound, $K$, we hypothesize that our lower bound can be leveraged to streamline the search across latent dimensions, $l$, such that the threshold, $\tau$, is satisfied, shrinking the search space *before any training*. More formally, let $\mathbb{L} = \{l_1, \ldots, l_Q\}$, denote the NAS search space over $l$, which is ascending. For each $l_i \in \mathbb{L}$, we compute $\hat{\mathcal{F}}_{\eta_i}^*$, using Theorem 4.6, for $\eta_i = (d, l_i, K, h_{\mathcal{D}})$. When $\tau > \hat{\mathcal{F}}_{\eta_i}^*$, we train an autoencoder with latent dimension $l_i$, terminating the search process when the validation error satisfies $\tau$ (see Algorithm 1).

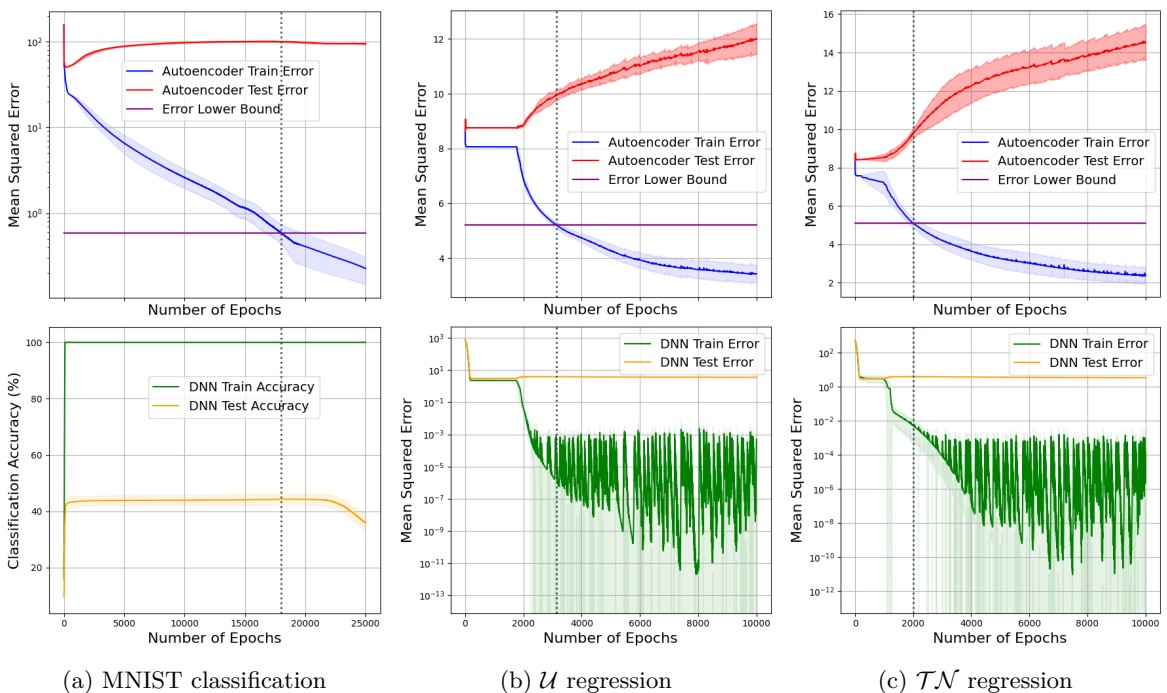

Figure 3: Overfitting detection results for MNIST classification, uniform distribution regression, and truncated normal distribution regression. The bottom row refers to the network, $t(f(\cdot))$, and the top row refers to the corresponding autoencoder, $g(f(\cdot))$. The x-axis denotes the number of elapsed training epochs, while the y-axis indicates the mean squared error.

We now consider the synthetic $\mathcal{U}$ and $\mathcal{TN}$ datasets from Section 7.1.1, such that $d = 100$, $K \approx 0.1$, $L = 2$, $|\mathcal{I}| = 5000$, and $|\mathcal{I}_{\text{val}}| = 10,000$, with $h_{\mathcal{D}} = 0$ for the uniform distribution case, and $h_{\mathcal{D}} \approx -1.036$ for the truncated normal case. We define a search set, $\mathbb{L} = \{1, 2, 4, 10, 20, 40, 46, 48\}$, and consider threshold $\tau = 4.5$ for illustration purposes, performing NAS to choose the architecture with smallest $l_i \in \mathbb{L}$ satisfying $\tau$. Table 1 summarizes the NAS result, where our lower bound informs the reduced search space, $l_i \in \mathbb{L} \setminus \{1, 2, 4\}$, with $_{0.1}\mathcal{A}_{100 \mapsto 46 \mapsto 100}$ being selected through Algorithm 1.

---

**Algorithm 1:** Neural Architecture Search with $\hat{\mathcal{F}}_\eta^*$

---

**Input:** $d$, $h_{\mathcal{D}}$, $K$, $\tau$, $\mathbb{L}$, $\mathcal{I}$, $\mathcal{I}_{\text{val}}$
**Output:** $l_i$, $\mathcal{A}_\theta$ (Trained autoencoder)

1   **for** $l_i \in \mathbb{L}$ **do**
2     Compute $\hat{\mathcal{F}}_{\eta_i}^*$ for $\eta_i = (d, l_i, h_{\mathcal{D}}, K)$
3     **if** $\tau > \hat{\mathcal{F}}_{\eta_i}^*$ **then**
4       $\mathcal{A} \leftarrow {}_K\mathcal{A}_{d \mapsto l_i \mapsto d}$ (Initialize autoencoder)
5       $\theta \leftarrow \theta_0$ (Initialize model parameters)
6       $\theta \leftarrow \arg\min_\theta \mathcal{L}(\mathcal{A}_\theta; \mathcal{I})$ (Train autoencoder)
7       $Err_{\text{val}} \leftarrow \frac{1}{|\mathcal{I}_{\text{val}}|} \sum_{x \in \mathcal{I}_{\text{val}}} \|x - \mathcal{A}_\theta(x)\|^2$
8       **if** $Err_{val} \leq \tau$ **then**
9        **return** $l_i, \mathcal{A}_\theta$
10     **else**
11       **skip** $l_i$
12 **return** *No $\mathcal{A}_\theta$ found satisfying $\tau$*

---

Table 1: Neural architecture search with $\hat{\mathcal{F}}_\eta^*$ on $\mathcal{U}$ reconstruction and $\mathcal{TN}$ reconstruction.

| $l_i \in \mathbb{L}$ | $l_i = 1$ | | $l_i = 2$ | | $l_i = 4$ | | $l_i = 10$ | |
|---|---|---|---|---|---|---|---|---|
| **Dataset** | **MSE** | $\hat{\mathcal{F}}_\eta^*$ | **MSE** | $\hat{\mathcal{F}}_\eta^*$ | **MSE** | $\hat{\mathcal{F}}_\eta^*$ | **MSE** | $\hat{\mathcal{F}}_\eta^*$ |
| $\mathcal{U}$ | $8.343_{\pm 0.012}$ | $5.4913$ | $8.328_{\pm 0.010}$ | $5.2123$ | $8.225_{\pm 0.010}$ | $4.7388$ | $7.661_{\pm 0.006}$ | $3.6165$ |
| $\mathcal{TN}$ | $7.989_{\pm 0.010}$ | $5.3765$ | $7.972_{\pm 0.016}$ | $5.1011$ | $7.886_{\pm 0.018}$ | $4.6336$ | $7.324_{\pm 0.005}$ | $3.5257$ |
| $l_i \in \mathbb{L}$ | $l_i = 20$ | | $l_i = 40$ | | $l_i = 46$ | | $l_i = 48$ | |
| **Dataset** | **MSE** | $\hat{\mathcal{F}}_\eta^*$ | **MSE** | $\hat{\mathcal{F}}_\eta^*$ | **MSE** | $\hat{\mathcal{F}}_\eta^*$ | **MSE** | $\hat{\mathcal{F}}_\eta^*$ |
| $\mathcal{U}$ | $6.719_{\pm 0.010}$ | $2.1986$ | $4.976_{\pm 0.004}$ | $0.3174$ | $\mathbf{4.456_{\pm 0.002}}$ | $\mathbf{0.0321}$ | $4.285_{\pm 0.004}$ | $0.0015$ |
| $\mathcal{TN}$ | $6.417_{\pm 0.008}$ | $2.1289$ | $4.771_{\pm 0.003}$ | $0.2916$ | $\mathbf{4.268_{\pm 0.003}}$ | $\mathbf{0.0250}$ | $4.104_{\pm 0.004}$ | $0.0009$ |

## 8 Limitations

While our proposed lower bound is a valid bound on the generalization MSE of autoencoders, we note that Theorem 4.5 demonstrates the most informative bound is strictly positive when $l < d/2$ under reasonable constraints. As we discuss in Section F of the Appendix, however, the generalization MSE should be strictly positive $\forall l < d$. Thus, our bound loses tightness in the regime of $l \in [d/2, d)$. Moreover, while estimating the differential entropy, $h_{\mathcal{D}}$, is not central to our work, we note that current estimation methods often observe diminished performance in high dimensional settings.

## 9 Conclusion

In this paper, we presented an information-theoretic lower bound the generalization MSE of autoencoders with sigmoid activation functions. The lower bound is a function of the data distribution and the autoencoder architecture, and is an objective condition that cannot be violated with infinite data and perfect training. We analyzed properties of this bound and matched these properties to intuition. We applied this bound to streamline neural architecture search and to detect overfitting on synthetic and benchmark-image datasets.

### Acknowledgments

The authors would like to thank Prof. Galen Reeves for insightful discussions relating to the development of the bound. Shyam Venkatasubramanian and Vahid Tarokh were supported in part by the Air Force Office of Scientific Research under award FA9550-21-1-0235.

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

# Appendix

This Appendix is divided into five main sections, one for each respective section from the main text, followed by a discussion of the impossibility of a continuous bijective and compressive transformation in Section F.

## A    Lower Bound from Injective Noise

### A.1    Lower Bound on Term 2 (Difference Between Noisy and Noiseless Decoders)

*Proof of Remark 3.2.* We begin by forming a lower bound on the mean squared difference of the reconstruction between the noisy and noiseless decoders. By the properties of the norm, we have that:

$$\mathbb{E}\left[\|g(f(X)) - g(f(X) + Z)\|^2\right] \geq 0. \tag{25}$$

$\square$

### A.2    Lower Bound on Term 3 (Noisy Autoencoder)

*Proof of Lemma 3.3.* Subsequently, we derive a lower bound on the MSE of the noise-injected autoencoder. We recall the estimation error and differential entropy (EEDE) inequality in the context of our noise-injected autoencoder:

$$\mathbb{E}\left[(X - g(f(X) + Z))^2\right] = \mathbb{E}\left[(X - g(Y + Z))^2\right] \geq \frac{1}{2\pi e} \exp(2h(X|Y + Z)). \tag{26}$$

As this inequality pertains to the scalar case, we extend it to the case where $X, g(f(X)+Z)$ are $d$-dimensional random vectors. Consider the $l$-dimensional vector $W = Y+Z$, wherein the covariance matrix of $(X|W = \mathrm{w})$, $\mathbb{E}\left[((X|W = \mathrm{w}) - \mu_{X|W=\mathrm{w}})((X|W = \mathrm{w}) - \mu_{X|W=\mathrm{w}})^T\right]$, is positive semi-definite. Accordingly, we have that:

$$\mathbb{E}\left[\|X - g(W)\|^2 \mid W = \mathrm{w}\right] \geq \mathbb{E}\left[\|X - \mu_{X|W=w}\|^2 \mid W = \mathrm{w}\right]$$

$$= \mathrm{tr}\left(\mathbb{E}\left[((X|W = \mathrm{w}) - \mu_{X|W=\mathrm{w}})((X|W = \mathrm{w}) - \mu_{X|W=\mathrm{w}})^T\right]\right) \tag{27}$$

$$\geq d\left[\det\left(\mathbb{E}\left[((X|W = \mathrm{w}) - \mu_{X|W=\mathrm{w}})((X|W = \mathrm{w}) - \mu_{X|W=\mathrm{w}})^T\right]\right)\right]^{\frac{1}{d}}. \tag{28}$$

This follows from the orthogonality principle, as $\mathbb{E}[\|X-g(W)\|^2 \mid W = \mathrm{w}]$ is minimized when $g(w) = \mu_{X|W=w}$. We also recall the conditional differential entropy, $h(X|W = \mathrm{w})$, is maximized for a $d$-dimensional Gaussian distribution, where:

$$h(X|W = \mathrm{w}) = \frac{d}{2}\log\left(2\pi e\left[\det\left(\mathbb{E}\left[((X|W = \mathrm{w}) - \mu_{X|W=\mathrm{w}})((X|W = \mathrm{w}) - \mu_{X|W=\mathrm{w}})^T\right]\right)\right]^{\frac{1}{d}}\right). \tag{29}$$

Rearranging this expression, we obtain:

$$\frac{1}{2\pi e}\exp\left(\frac{2}{d}h(X|W = \mathrm{w})\right) = \left[\det\left(\mathbb{E}\left[((X|W = \mathrm{w}) - \mu_{X|w=\mathrm{w}})((X|w = \mathrm{w}) - \mu_{X|W=\mathrm{w}})^T\right]\right)\right]^{\frac{1}{d}}. \tag{30}$$

Therefore, per Eq. (27), we have that:

$$\forall \mathrm{w} \in \mathbb{R}^l, \ \mathbb{E}\left[\|X - g(W)\|^2 \mid W = \mathrm{w}\right] \geq \frac{d}{2\pi e}\exp\left(\frac{2}{d}h(X|W = \mathrm{w})\right) \tag{31}$$

$$\mathbb{E}\left[\|X - g(Y + Z)\|^2\right] = \mathbb{E}\left[\|X - g(W)\|^2\right] \geq \frac{d}{2\pi e}\exp\left(\frac{2}{d}h(X|W)\right) = \frac{d}{2\pi e}\exp\left(\frac{2}{d}h(X|Y + Z)\right). \tag{32}$$

To complete the derivation, we must obtain a lower bound on $h(X|Y+Z)$. We begin by recall that:

$$h(X|Y+Z) = h_{\mathcal{D}} - h(Y+Z) + h(Y+Z|X). \tag{33}$$

Consider $h(Y+Z)$, where $Y$ and $Z$ are independent. We recall that $Y$ and $Z$ have finite covariance matrices, $\mathbb{E}\big[(Y-\mu_Y)(Y-\mu_Y)^T\big]$ and $\sigma^2 I_l$, respectively, with $\mu_Y$ denoting the mean of $Y$. As $Y$ denotes the post-sigmoid value, it exists on the support of $[0,1]^l$. Accordingly, the diagonal elements of $\mathbb{E}\big[(Y-\mu_Y)(Y-\mu_Y)^T\big]$ cannot exceed the variance of a Bernoulli distribution with $p = 0.5$. It now follows that:

$$\begin{aligned}
h(Y+Z) &\leq \frac{1}{2}\log\left((2\pi e)^l\big[\det\big(\mathbb{E}[(Y-\mu_Y)(Y-\mu_Y)^T] + \mathbb{E}[ZZ^T]\big)\big]\right) \\
&\leq \frac{1}{2}\log\left((2\pi e)^l\left[\det\left(\frac{1}{4}I_l + \sigma^2 I_l\right)\right]\right) \\
&= \frac{l}{2}\log\left(2\pi e\left(\frac{1}{4} + \sigma^2\right)\right).
\end{aligned} \tag{34}$$

We also know that $h(Y+Z|X) = h(f(X)+Z|X) = h(Z)$, as $Z$ is independent of $X$. Putting it all together:

$$h(X|Y+Z) \geq h_{\mathcal{D}} - \frac{l}{2}\log\left(2\pi e\left(\frac{1}{4} + \sigma^2\right)\right) + \frac{l}{2}\log(2\pi e\sigma^2). \tag{35}$$

Cumulatively, we have that:

$$\mathbb{E}\left[\|X - g(Y+Z)\|^2\right] \geq \frac{d}{2\pi e}\exp\left(\frac{2}{d}\left[h_{\mathcal{D}} - \frac{l}{2}\log\left(2\pi e\left(\frac{1}{4} + \sigma^2\right)\right) + \frac{l}{2}\log(2\pi e\sigma^2)\right]\right). \tag{36}$$

$\square$

### A.3 Upper Bound on Term 4 (Cross-Term)

*Proof of Lemma 3.4.* We now form an upper bound on the cross-term through the absolute value function, and apply Jensen's inequality and the Cauchy-Schwarz inequality for random vectors and random variables:

$$\begin{aligned}
\mathbb{E}\left[2[g(f(X)) - g(f(X)+Z)]^T[X - g(f(X)+Z)]\right] &\leq \left|\mathbb{E}\left[2[g(f(X)) - g(f(X)+Z)]^T[X - g(f(X)+Z)]\right]\right| \\
&\leq 2\mathbb{E}\left[\left|[g(f(X)) - g(f(X)+Z)]^T[X - g(f(X)+Z)]\right|\right] \\
&\leq 2\mathbb{E}\left[\|g(f(X)) - g(f(X)+Z)\|\|X - g(f(X)+Z)\|\right] \\
&\leq 2\sqrt{\mathbb{E}\big[\|g(f(X)) - g(f(X)+Z)\|^2\big]\mathbb{E}\big[\|X - g(f(X)+Z)\|^2\big]}
\end{aligned} \tag{37}$$

We observe that the square root function comprises the MSE of the noise-injected autoencoder and the mean squared difference between the noisy and noiseless decoders. We form an upper bound on both terms to form an upper bound on the cross-term. We begin by forming an upper bound on the MSE of the noise-injected autoencoder. We know that the decoder, $g$, defines a Lipschitz function, where $K^2$ is the squared Lipschitz constant upper bound:

$$\forall x \in \mathcal{X}, z \in \mathbb{R}^l, \ \|g(f(x)) - g(f(x)+z)\|^2 \leq K^2\|f(x) - (f(x)+z)\|^2 = K^2\|z\|^2. \tag{38}$$

As $Z \sim \mathcal{N}(0, \sigma^2 I_l)$, then $\mathbb{E}\big[\|Z\|^2\big] = l\sigma^2$. Accordingly, we have that:

$$\mathbb{E}\left[\|g(f(X)) - g(f(X)+Z)\|^2\right] \leq K^2 l\sigma^2. \tag{39}$$

We now form an upper bound on the mean squared difference between the noisy and noiseless decoders. Since $X, g(f(X) + Z) \in [0,1]^d$, the vector corresponding to the maximum difference is given by a $d$-dimensional vector of ones, $\mathbf{1}_d$. Thus, it follows that:

$$\mathbb{E}\left[\|X - g(f(X) + Z)\|^2\right] \leq \sup_{X,Z} \mathbb{E}\left[\|X - g(f(X) + Z)\|^2\right] \leq \|\mathbf{1}_d\|^2 = d. \tag{40}$$

Therefore, we have that:

$$\mathbb{E}\left[2[g(f(X)) - g(f(X) + Z)]^T[X - g(f(X) + Z)]\right] \leq 2\sqrt{(K^2 l \sigma^2)(d)} = 2K\sqrt{l d \sigma^2}. \tag{41}$$

$\square$

### A.4 Lower Bound on Term 1 (Noiseless Autoencoder)

*Proof of Theorem 3.5.* We have now bounded each of the terms in Eq. (5). As such, to obtain a lower bound on the MSE of the noiseless autoencoder, we substitute these bounds for the terms in Eq. (5):

$$\mathbb{E}\left[\|X - g(f(X))\|^2\right] = \mathbb{E}\left[\|X - g(f(X) + Z)\|^2\right] + \mathbb{E}\left[\|g(f(X)) - g(f(X) + Z)\|^2\right]$$
$$- \mathbb{E}\left[2[g(f(X)) - g(f(X) + Z)]^T[X - g(f(X) + Z)]\right]$$
$$\geq \frac{d}{2\pi e}\exp\left(\frac{2}{d}\left[h_{\mathcal{D}} - \frac{l}{2}\log\left(2\pi e\left(\frac{1}{4} + \sigma^2\right)\right) + \frac{l}{2}\log(2\pi e \sigma^2)\right]\right) - 2K\sqrt{l d \sigma^2}. \tag{42}$$

We note that this expression can be significantly simplified. By consolidating similar terms, we obtain:

$$\mathbb{E}\left[\|X - g(f(X))\|^2\right] \geq \frac{d}{2\pi e}\exp\left(\frac{2}{d}\left[h_{\mathcal{D}} - \frac{l}{2}\log\left(2\pi e\left(\frac{1}{4} + \sigma^2\right)\right) + \frac{l}{2}\log(2\pi e \sigma^2)\right]\right) - 2K\sqrt{l d \sigma^2}$$
$$= \frac{d}{2\pi e}\exp\left(\frac{2h_{\mathcal{D}}}{d}\right)\exp\left(\left(\frac{-l}{d}\right)\log\left(2\pi e\left(\frac{1}{4} + \sigma^2\right)\right)\right)\exp\left(\frac{l}{d}\log(2\pi e \sigma^2)\right) - 2K\sqrt{l d \sigma^2}$$
$$= \frac{d}{2\pi e}\exp\left(\frac{2h_{\mathcal{D}}}{d}\right)\left(2\pi e\left(\frac{1}{4} + \sigma^2\right)\right)^{\frac{-l}{d}}(2\pi e \sigma^2)^{\frac{l}{d}} - 2K\sqrt{l d \sigma^2}$$
$$= \frac{d}{2\pi e}\exp\left(\frac{2h_{\mathcal{D}}}{d}\right)(2\pi e)^{-\frac{l}{d}}\left(\frac{1}{4} + \sigma^2\right)^{\frac{-l}{d}}(2\pi e)^{\frac{l}{d}}(\sigma^2)^{\frac{l}{d}} - 2K\sqrt{l d \sigma^2}$$
$$= \frac{d}{2\pi e}\exp\left(\frac{2h_{\mathcal{D}}}{d}\right)\left(\frac{\sigma^2}{\frac{1}{4} + \sigma^2}\right)^{\frac{l}{d}} - 2K\sqrt{l d \sigma^2}. \tag{43}$$

For brevity, we define:

$$s \triangleq \sigma^2, \qquad \alpha \triangleq \frac{d}{2\pi e}, \qquad \beta \triangleq \exp\left(\frac{2h_{\mathcal{D}}}{d}\right), \qquad \text{and} \qquad \gamma \triangleq 2K\sqrt{l d}. \tag{44}$$

Therefore, the lower bound on the generalization MSE of the noiseless autoencoder is given by:

$$\mathcal{F}(s, d, l, K, h_{\mathcal{D}}) = \alpha\beta\left(\frac{s}{\frac{1}{4} + s}\right)^{\frac{l}{d}} - \gamma\sqrt{s}. \tag{45}$$

$\square$

# B  Properties of the Lower Bound

## B.1  Lower Bound Extrema

*Proof of Lemma 4.1.* We prove the three claims separately:

1. We first show that $\mathcal{F}_\eta(0) = 0$:

$$\mathcal{F}_\eta(0) = \alpha\beta(0)^{\frac{l}{d}} - \gamma\sqrt{0} = 0. \tag{46}$$

2. We next show that $\lim_{s\to\infty} \mathcal{F}_\eta(s) = -\infty$. Accordingly, we consider Eq. (45), and again refer to the difference and product laws for limits, alongside the power law for limits:

$$\lim_{s\to\infty} \mathcal{F}_\eta(s) = \lim_{s\to\infty} \left[ \alpha\beta \left( \frac{s}{\frac{1}{4}+s} \right)^{\frac{l}{d}} - \gamma\sqrt{s} \right]$$

$$= \alpha\beta \lim_{s\to\infty} \left[ \left( \frac{s}{\frac{1}{4}+s} \right)^{\frac{l}{d}} \right] - \lim_{s\to\infty} \left[ \gamma\sqrt{s} \right]$$

$$= \alpha\beta \lim_{s\to\infty} \left[ \left( \frac{s}{\frac{1}{4}+s} \right) \right]^{\frac{l}{d}} - [\infty]$$

$$= \alpha\beta[1]^{\frac{l}{d}} - [\infty] = -\infty.$$

Therefore, $\lim_{s\to\infty} \mathcal{F}_\eta(s) = -\infty$.

3. Finally, we show that, for all $s$, $\mathcal{F}_\eta(s) \leq d$. We note that, for all $s, l, d > 0$:

$$\left( \frac{s}{\frac{1}{4}+s} \right)^{\frac{l}{d}} \leq 1. \tag{47}$$

Next, we note that as $\mathcal{D}$ is supported by $\mathcal{X} = [0,1]^d$, it follows that $h_\mathcal{D} \leq 0$ and that $0 \leq \beta \leq 1$. We finally observe that $\gamma \geq 0$. Putting these facts together, we get that:

$$\alpha\beta \left( \frac{s}{\frac{1}{4}+s} \right)^{\frac{l}{d}} - \gamma\sqrt{s} \leq \alpha. \tag{48}$$

The result follows as $\alpha < d$, from the definition of $\alpha$.

$\square$

## B.2  Lower Bound Dependencies

In this section, we obtain the pointwise derivatives of $\mathcal{F}_\eta(s)$ with respect to its parameters, $\eta = (d, l, K, h_\mathcal{D})$. We assume $s \in (0, \infty)$, as this is the domain of interest. We emphasize that these dependencies are NOT specifically with respect to $s^*$, the optimal noise power level.

*Proof of Lemma 4.2.* We take the derivative analytically with respect to $K$ and $h_\mathcal{D}$:

$$\frac{\partial \mathcal{F}_\eta(s)}{\partial K} = -2\sqrt{lds} < 0, \tag{49}$$

$$\frac{\partial \mathcal{F}_\eta(s)}{\partial h_\mathcal{D}} = \alpha\beta \left( \frac{s}{\frac{1}{4}+s} \right)^{\frac{l}{d}} \left( \frac{2}{d} \right) > 0. \tag{50}$$

Although $l$ is defined to be a positive integer, we consider a continuous extension of $l$, which we treat as a continuous variable. If the derivative of this function is strictly negative, then $\mathcal{F}_\eta(s)$ is a decreasing function in integer $l$:

$$\frac{\partial \mathcal{F}_\eta(s)}{\partial l} = \alpha\beta\left(\frac{s}{\frac{1}{4}+s}\right)^{\frac{l}{d}}\left(\frac{1}{d}\right)\log\left(\frac{s}{\frac{1}{4}+s}\right) - K\sqrt{\frac{ds}{l}} < 0. \tag{51}$$

This result follows from the facts that *(i)* the first term is negative as $s < \frac{1}{4} + s$, and *(ii)* $K\sqrt{\frac{ds}{l}} > 0$. $\qquad\square$

While no claim was made regarding the dependence of $\mathcal{F}_\eta(s)$ on $d$, we nonetheless offer the following derivative of a continuous extension of discrete integer $d$:

$$\frac{\partial \mathcal{F}_\eta(s)}{\partial d} = \frac{1}{2\pi e}\exp\left(\frac{2h_\mathcal{D} + l\log\left(\frac{\sigma^2}{\frac{1}{4}+\sigma^2}\right)}{d}\right)\left(1 - \frac{2h_\mathcal{D} + l\log\left(\frac{\sigma^2}{\frac{1}{4}+\sigma^2}\right)}{d}\right) - \frac{K\sqrt{l}\,\sigma^2}{\sqrt{d}}, \tag{52}$$

This dependence is more complex. Practically, we observe that for sufficiently small $d$, $\frac{\partial \mathcal{F}_\eta(s)}{\partial d} < 0$, and for sufficiently large $d$, $\frac{\partial \mathcal{F}_\eta(s)}{\partial d} > 0$. This derivative is not uniformly greater than or less than zero.

### B.3 Lower Bound Nonnegativity

We first introduce a function, $r(s)$, that multiple proofs in this section will leverage as a proof technique:

$$r(s) \overset{\Delta}{=} \left(\frac{s}{s + \frac{1}{4}}\right)^{-\frac{l}{d}}\sqrt{s}. \tag{53}$$

Before proving Theorem 4.5, we must first prove the following four Lemmas:

**Lemma B.1.**

$$\mathcal{F}_\eta(s) > 0 \iff \frac{\alpha\beta}{\gamma} > r(s). \tag{54}$$

*Proof.* Recall the definitions of $\alpha, \beta, \gamma$ in Eq. (44). We refer to the definition of $\mathcal{F}_\eta(s)$ from Eq. (45). Clearly:

$$\alpha\beta\left(\frac{s}{\frac{1}{4}+s}\right)^{\frac{l}{d}} - \gamma\sqrt{s} > 0 \iff \frac{\alpha\beta}{\gamma} > \left(\frac{s}{\frac{1}{4}+s}\right)^{-\frac{l}{d}}\sqrt{s} = r(s). \tag{55}$$

$\qquad\square$

**Remark B.2.** *We note that $(\alpha\beta)/\gamma$ is constant in $s$.*

**Lemma B.3.** *Under Assumption 4.3, if $l < \frac{d}{2}$, $\exists s_0 > 0$ such that $\mathcal{F}_\eta(s_0) > 0$.*

*Proof.* Under Assumption 4.3, $\alpha, \beta, \gamma > 0$, so $\alpha\beta/\gamma$ is a strictly positive quantity that is constant in $s$. For $0 \le s \le \frac{1}{4}$, we have that:

$$2s \le \frac{s}{\frac{1}{4}+s} \implies (2s)^{-\frac{l}{d}} \ge \left(\frac{s}{s+\frac{1}{4}}\right)^{-\frac{l}{d}}. \tag{56}$$

We now define that:

$$t(s) = (2s)^{-\frac{l}{d}}\sqrt{s} = (2^{-\frac{l}{d}})(s^{\frac{1}{2}-\frac{l}{d}}), \tag{57}$$

and observe that for $0 < s < \frac{1}{4}$, $0 \le r(s) \le t(s)$.

Thus, when $l < \frac{d}{2}$, $\frac{1}{2} - \frac{l}{d} > 0$, it follows that $\lim_{s\to 0^+} t(s) = 0$. Moreover, $r(s) \ge 0$ for $s \ge 0$, so by the squeeze theorem, we know that $\lim_{s\to 0^+} r(s) = 0$. Altogether, since $r(s) \to 0$ as $s \to 0^+$, we know $\exists s_0 > 0$ such that $(\alpha\beta)/\gamma > r(s_0)$, as the quantity on the left-hand side is strictly positive. $\qquad\square$

**Lemma B.4.** *Under Assumption 4.4, if $l \geq \frac{d}{2}$, then $\frac{\alpha\beta}{\gamma} \leq \frac{1}{2}$.*

*Proof.* We begin with Assumption 4.4:

$$K \geq \frac{1}{\pi e \sqrt{2}} \implies K\pi e \sqrt{2} \geq 1 \implies K^2 \pi^2 e^2 2 \geq 1 \implies K^2 \pi^2 e^2 4 \left(\frac{d}{2}\right) \geq d, \tag{58}$$

where the last step follows from multiplying by $d$, which we know is positive. We are also given $l \geq \frac{d}{2}$. Thus:

$$K^2 \pi^2 e^2 4(l) \geq d, \tag{59}$$

which implies that:

$$K^2 l \geq \frac{d}{4\pi^2 e^2} \implies K\sqrt{l} \geq \frac{\sqrt{d}}{2\pi e} \implies K\sqrt{ld} \geq \frac{d}{2\pi e}. \tag{60}$$

This expression is equivalent to:

$$\frac{\gamma}{2} \geq \alpha \implies \frac{1}{2} \geq \frac{\alpha}{\gamma}. \tag{61}$$

We note that, as $\mathcal{D}$ is supported on $\mathcal{X} = [0,1]^d$, we know that $h_{\mathcal{D}} \leq 0$. Thus, $0 \leq \beta \leq 1$, and:

$$\frac{\alpha}{\gamma} \geq \frac{\alpha\beta}{\gamma}. \tag{62}$$

Putting everything together, we arrive at:

$$\frac{\alpha\beta}{\gamma} \leq \frac{1}{2}. \tag{63}$$

$\square$

**Lemma B.5.** *Under Assumption 4.4, when $l \geq \frac{d}{2}$, $\mathcal{F}_\eta(s) \leq 0$ for all $s > 0$.*

*Proof.* We have previously calculated that $\mathcal{F}_\eta(0) = 0$. Correspondingly, we must show that for all $s > 0$, $\mathcal{F}_\eta(s) \leq 0$. When $l \leq \frac{d}{2}$, we have that $\forall s > 0$:

$$\left(\frac{s}{s+1/4}\right)^{-\frac{l}{d}} \geq \left(\frac{s}{s+1/4}\right)^{-\frac{1}{2}} \geq \left(\frac{s}{1/4}\right)^{-\frac{1}{2}}. \tag{64}$$

Recall $r(s)$ from Eq. (53). Suppose for contradiction there exists a value $s_0 > 0$ such that $r(s_0) < \frac{1}{2}$. Then:

$$r(s_0) = \left(\frac{s_0}{s_0 + 1/4}\right)^{-\frac{l}{d}} \sqrt{s_0} < \frac{1}{2} \iff \left(\frac{s_0}{s_0 + 1/4}\right)^{-\frac{l}{d}} < \frac{1}{2} s_0^{-\frac{1}{2}} = \left(\frac{s_0}{1/4}\right)^{-\frac{1}{2}},$$

which contradicts Eq. (64). Thus, for all $s > 0$, $r(s) \geq \frac{1}{2}$ when $l \geq \frac{d}{2}$. As we assume Assumption 4.4, we can use the result of Lemma B.4: $\alpha\beta/\gamma \leq \frac{1}{2}$. Finally, we have that:

$$\frac{\alpha\beta}{\gamma} \leq \frac{1}{2} < r(s), \tag{65}$$

for all $s > 0$ when $l \geq \frac{d}{2}$, which means that $\mathcal{F}_\eta(s) \leq 0$ for all $s > 0$ by Lemma B.1. $\square$

We are now finally prepared to prove Theorem 4.5:

*Proof of Theorem 4.5.* As we assume Assumptions 4.3 and 4.4, we can leverage Lemmas B.3 and B.5. We further note that if $\exists s_0$ such that $\mathcal{F}_\eta(s_0) > 0$, then clearly $\mathcal{F}_\eta(s^*) > 0$ as $\mathcal{F}_\eta(s^*) \geq \mathcal{F}_\eta(s_0)$. Then:

$$l < \frac{d}{2} \iff \mathcal{F}_\eta^* > 0. \tag{66}$$

$\square$

### B.4 The *Impractical* Regime

For completeness, we now analyze the lower bound function, $\mathcal{F}_\eta(s)$, within the "impractical" regime, where $h_{\mathcal{D}} = -\infty$ or $K \leq w(d, l, h_{\mathcal{D}})$, where $w$ is a function with range $[0, \frac{1}{\pi e \sqrt{2}}]$. In this regime, $\mathcal{F}_\eta(s)$ observes different qualitative behavior than in the previous case. We delineate this behavior in Sections B.4.1 and B.4.2, and we leverage the result of Lemma B.1.

#### B.4.1 Degenerate or Countable Data-Generating Distribution

If $h_{\mathcal{D}} = -\infty$ (e.g., the distribution of $x$ is degenerate or is of countable support in the discrete case), then $\beta = 0$, and thus, $\alpha\beta/\gamma = 0$. Accordingly, $r(s) \geq \alpha\beta/\gamma$ for all $s$, meaning that $\nexists s_0 : \mathcal{F}_\eta(s_0) > 0$. This observation aligns with theoretical expectations: if the distribution of $x$ is degenerate, it is consistent for the decoder to learn a constant function that yields zero MSE. Even if $x$ has countable support, the latent layer of dimension $l = 1$ has uncountable support. This allows the encoder to be injective, theoretically enabling perfect reconstruction, as each distinct input can map uniquely in the latent space.

#### B.4.2 Exceedingly Small Lipschitz Constant

When $K \leq w(d, l, h_{\mathcal{D}})$, wherein $w$ is a function with range given by $[0, \frac{1}{\pi e \sqrt{2}}]$ (see the proof of Lemma B.4), we empirically observe that the lower bound, $\mathcal{F}_\eta(s)$, begins at the origin and descends, before rising into the positive region and again descending to $-\infty$, yielding two critical points. This behavior contrasts with that of the practical regime from Section 4 of the main text.

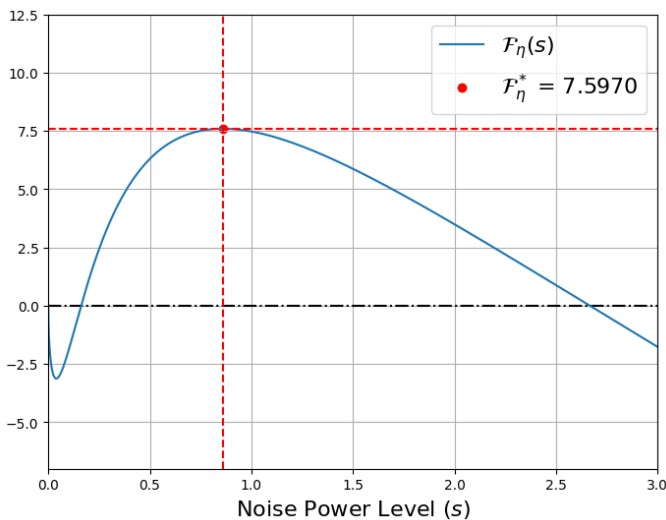

Figure 4: MSE lower bound as a function of $s$ for $d = 784, l = 1300, h_{\mathcal{D}} = 0$, and $K = 0.012$.

Deconstructing this behavior, for fixed $l, d \geq 1$ and $h_{\mathcal{D}} > -\infty$, we observe that when $K$ is extremely small, $\gamma$ is also small, and thus, $\frac{\alpha\beta}{\gamma}$ is exceedingly large. More concretely, we have that:

$$(K \to 0) \implies (\gamma \to 0) \implies \left(\frac{\alpha\beta}{\gamma} \to \infty\right). \tag{67}$$

Recall $r(s)$ from Eq. (53), and the relation from Lemma B.1. Consequently, it is possible that $r(s) < (\alpha\beta)/\gamma$ for $l \geq \frac{d}{2}$. Qualitatively, if $K$ is very small, this indicates that the decoder has severely limited expressivity, to such an extent that perfect reconstruction would be unattainable even in extreme cases where $l \geq d$. This is not due to information loss during encoding, but rather is due the decoder's profound lack of expressiveness. As such, a nonzero reconstruction MSE lower bound is possible even when $l \geq d/2$.

We recall that in the practical regime, $s^*$ is extremely small and close to zero. For the case $K \leq w(d, l, h_\mathcal{D})$, however, $s^*$ is no longer close to zero. This is intuitive, as previously, the factor driving $s^*$ towards zero was the magnitude of $K$; a larger $K$ places a greater penalty on higher values of $s$ (deriving from the pronounced disparity between the noisy and noiseless autoencoders in such scenarios). Conversely, when $K$ is smaller, this penalty on higher values of $s$ diminishes, removing the constraint that previously kept $s^*$ near zero.

### B.5 Estimating the Most Informative Bound

*Proof of Theorem 4.6.* We now attempt to analytically determine an estimate, $\hat{s}^*$, of $s^*$, with the approximation of Eq. (14). We rewrite the lower bound function from Eq. (45) using this approximation:

$$\hat{\mathcal{F}}_\eta(s) = \alpha\beta(4s)^{\frac{l}{d}} - \gamma\sqrt{s}. \tag{68}$$

Taking the derivative of $\hat{\mathcal{F}}_\eta(s)$, with respect to $s$, and setting it equal to zero, we obtain:

$$\frac{\partial \hat{\mathcal{F}}_\eta(s)}{\partial s} = 0 \implies \alpha\beta 4^{\frac{l}{d}}\left(\frac{l}{d}\right)s^{\frac{l}{d}-1} - \gamma\left(\frac{1}{2}\right)s^{-\frac{1}{2}} = 0$$

$$\iff \frac{\alpha\beta l}{d}4^{\frac{l}{d}}s^{\frac{l}{d}-1} = \frac{\gamma}{2}s^{-\frac{1}{2}}$$

$$\iff s^{\frac{l}{d}-\frac{1}{2}} = \frac{\gamma d}{2\alpha\beta 4^{\frac{l}{d}}l} \tag{69}$$

$$\iff s^{\frac{2l-d}{2d}} = \frac{\gamma d}{2\alpha\beta 4^{\frac{l}{d}}l} \tag{70}$$

$$\iff s = \left(\frac{\gamma d}{2\alpha\beta 4^{\frac{l}{d}}l}\right)^{\frac{2d}{2l-d}}. \tag{71}$$

The approximation $\hat{\mathcal{F}}_\eta^*$ can be found by applying the function $\mathcal{F}_\eta$ to $\hat{s}^*$. $\qquad\square$

## C Estimation of Bound Parameters

### C.1 Lower Bound Insensitivity to Overestimates of the Lipschitz Constant

We now establish an upper bound for $K_\theta$. We first start by showing that a *loose* upper bound on $K_\theta$ does not significantly compromise the tightness of $\mathcal{F}_\eta^*$, even when considering large changes in $K_\theta$.

*Proof of Theorem 5.1.* Let $s > 0$ and let $l < \frac{d}{2}$, we begin by recalling $r(s)$ from Eq. (53):

$$r(s) \triangleq \left(\frac{s}{s+\frac{1}{4}}\right)^{-\frac{l}{d}}\sqrt{s}. \tag{72}$$

Rewriting $r(s)$, we obtain:

$$r(s) = \left(\frac{s}{s+1/4}\right)^{-\frac{l}{d}}\sqrt{s} = \underbrace{\left(\frac{1}{s+1/4}\right)^{-\frac{l}{d}}}_{r_A(s)}\underbrace{s^{\frac{1}{2}-\frac{l}{d}}}_{r_B(s)}. \tag{73}$$

Clearly, both $r_A(s)$ and $r_B(s)$ are nonnegative for $s > 0$ and are monotonically increasing (assuming that $l < \frac{d}{2}$). Thus, $r(s)$ is nonnegative and monotonically increasing for all $s > 0$.

As $K_\theta$ approaches infinity, it is clear that $\alpha\beta/\gamma$ approaches zero. By Lemma B.1, we have that:

$$\alpha\beta/\gamma > r(s) \iff \mathcal{F}_\eta^* > 0. \tag{74}$$

Since (1) $\mathcal{F}_\eta(0) = 0$ and $\lim_{s\to\infty} \mathcal{F}_\eta(s) = -\infty$ by Lemma 4.1, (2) $\mathcal{F}_\eta(s)$ is continuous in $s$, and (3) $\exists s_0 > 0$ such that $\mathcal{F}_\eta(s_0) > 0$ when $l < \frac{d}{2}$ by Theorem 4.5, we can conclude that $\mathcal{F}_\eta(s)$ has a root $s^\dagger > 0 : \mathcal{F}_\eta(s^\dagger) = 0$. By Lemma B.1:

$$r(s^\dagger) = \alpha\beta/\gamma. \tag{75}$$

Since $r$ is monotonically increasing in $s$, we know the following, where $s^\dagger$ denotes the root of $\mathcal{F}_\eta(s)$:

$$\lim_{K\to\infty} \frac{\alpha\beta}{\gamma} = 0 \implies \lim_{K\to\infty} s^\dagger = 0. \tag{76}$$

As the optimal noise power level, $s^*$, is always less than or equal to $s^\dagger$, we have that: $0 < s^* \le s^\dagger$. Thus:

$$\lim_{K\to\infty} s^\dagger = 0 \implies \lim_{K\to\infty} s^* = 0. \tag{77}$$

Therefore, for large $K_\theta$, $s^*$ gets very close to zero. Recalling Eq. (49), we observe that the pointwise derivative of $\mathcal{F}_\eta(s)$ with respect to $K_\theta$ is $-2\sqrt{lds}$. As $s^*$ also depends on $K_\theta$, it follows that $|\frac{\partial \mathcal{F}_\eta^*}{\partial K_\theta}| < |-2\sqrt{lds^*}|$. Thus, the insensitivity of $\mathcal{F}_\eta^*(s)$ to large changes in $K_\theta$ stems from $-2\sqrt{lds^*}$ being small (as $s^*$ is arbitrarily small).

$\square$

## C.2 Upper Bounding the Lipschitz Constant

To compute an upper bound on the Lipschitz constant across various classes of functions, $\mathcal{H}$, for the decoder, $g$, we assume all of our activation functions are sigmoidal. We recall the Fundamental Theorem of Calculus via the notation of this section in Remark C.1:

**Remark C.1.** *Let $g : \mathbb{R}^l \mapsto \mathbb{R}^d$ be a continuously differentiable function with Jacobian matrix $Dg$. Letting $k(t) = g(y + th)$ for $t \in [0, 1]$, observe that $k'(t) = Dg(y + th)h$. Integrating both sides and applying the Fundamental Theorem of Calculus, we have that:*

$$g(y + h) - g(y) = \left(\int_0^1 Dg(y + th)dt\right) h \tag{78}$$

*Proof of Theorem 5.2.* Suppose $g : [0, 1]^l \to [0, 1]^d$, $y \mapsto \phi(\Theta_L \cdots \phi(\Theta_2 \phi(\Theta_1 y)))$, where $\phi$ is the component-wise sigmoid function. The sigmoid function is given by $\phi(z) = \frac{1}{1+e^{-z}}, \forall z \in \mathbb{R}$. In this context, $L - 1$ is the number of hidden layers in the decoder. For each layer, $k \in \{1, \ldots, L\}$, $\Theta_k$ belongs to $\mathcal{M}_{d_k \times d_{k+1}}(\mathbb{R})$, where $d_1 = l$ (latent space dimension) and $d_L = d$ (input space dimension). We assume $\forall k \in \{1, \ldots, L\}, d_k \le d$, which is a standard constraint for decoders. By Remark C.1, we now have that for $h = (h_1, \ldots, h_l)^T$:

$$\|g(y + h) - g(y)\|_2^2 = \left\|\left(\int_0^1 Dg(y + th)dt\right) h\right\|_2^2$$
$$= \sum_{j=1}^d \left(\sum_{i=1}^l h_i \int_0^1 \frac{\partial g_j}{\partial y_i}(y + th)dt\right)^2. \tag{79}$$

Hence, it follows from the Cauchy–Schwarz inequality that:

$$\|g(y + h) - g(y)\|_2^2 \le \sum_{j=1}^d \left(\sum_{i=1}^l h_i^2 \sum_{i=1}^l \left(\int_0^1 \frac{\partial g_j}{\partial y_i}(y + th)dt\right)^2\right)$$
$$= \sum_{i=1}^l h_i^2 \sum_{j=1}^d \sum_{i=1}^l \left(\int_0^1 \frac{\partial g_j}{\partial y_i}(y + th)dt\right)^2$$
$$= \|h\|_2^2 \|\Phi\|_F^2, \tag{80}$$

where $\Phi$ is the matrix that has $\Phi_{ij} = \int_0^1 \frac{\partial g_j}{\partial y_i}(y+th)dt$ as its entries. Thus, the problem is reduced to deriving a bound on the derivative of the decoder with respect to its input (the latent space encoded representation in the autoencoder). Suppose now that $L = 1$. We have that for $\theta_j$ (the $j^{th}$ line of the matrix $\Theta_1$):

$$\frac{\partial g_j}{\partial y_i}(y) = \frac{\partial}{\partial y_i}\phi(\theta_j^T y) = \theta_{ji}\phi(\theta_j^T y)\phi(1 - \theta_j^T y) \tag{81}$$

As the derivative of the sigmoid function is bounded above by $1/4$, we have that:

$$\left(\int_0^1 \frac{\partial g_j}{\partial y_i}(y+th)dt\right)^2 \leq \frac{\theta_{ji}^2}{4^2}. \tag{82}$$

Therefore, it follows that:

$$\|g(y+h) - g(y)\|_2^2 \leq \frac{1}{16}\|\Theta_1\|_F^2\|h\|_2^2. \tag{83}$$

When $L$ layers comprise the decoder, this generalizes to the following:

$$\|g(y+h) - g(y)\|_2^2 \leq \frac{\|h\|_2^2}{16^L} \prod_{k=1}^{L} \|\Theta_k\|_F^2. \tag{84}$$

We observe that upper bounding the norm, $\|\Theta_k\|_F$, is sufficient to establish an upper bound on the Lipschitz constant of the decoder. Therefore, under the maximum norm constraint, $M$, we have that:

$$\frac{\|g(y+h) - g(y)\|_2^2}{\|h\|_2^2} \leq K_\theta \leq \frac{M^{2L}}{16^L}, \quad \text{where:} \quad K_\theta = \frac{\prod_{k=1}^{L}\|\Theta_k\|_F^2}{16^L}. \tag{85}$$

Furthermore, we consider that for a function, $\phi(\Theta_i y)$, the Lipschitz constant remains of the same order when including a bias term, $b$, since the derivative of the sigmoid function with respect to the bias term is similarly bounded. Specifically, for each layer, $k$, the presence of the bias term, $b_k$, does not affect the overall Lipschitz constant by more than an additive constant, as the bias term appears linearly in the argument of $\phi$.

Let $g_b(y) = \phi(\Theta_1 y + b_1)$, where $b_1$ is the first layer's bias term. The derivative of $g_b(y)$ with respect to $y_i$ is:

$$\frac{\partial g_{b,j}}{\partial y_i}(y) = \frac{\partial}{\partial y_i}\phi(\theta_j^T y + b_j) = \theta_{ji}\phi(\theta_j^T y + b_j)\left(1 - \phi(\theta_j^T y + b_j)\right). \tag{86}$$

As the derivative of the sigmoid function is bounded by $1/4$, and the bias term does not introduce additional dependencies on $y$, we have that:

$$\left(\int_0^1 \frac{\partial g_{b,j}}{\partial y_i}(y+th)dt\right)^2 \leq \frac{\theta_{ji}^2}{4^2}. \tag{87}$$

Accordingly, the same upper bound applies in the presence of a bias term, leading to the following general bound for the decoder with bias terms included:

$$\|g_b(y+h) - g_b(y)\|_2^2 \leq \frac{\|h\|_2^2}{16^L} \prod_{k=1}^{L} \|\Theta_k\|_F^2. \tag{88}$$

Thus, the presence of bias terms does not affect the overall upper bound on the Lipschitz constant of the decoder. Under maximum norm regularization, the final bound remains:

$$\frac{\|g(y+h) - g(y)\|_2^2}{\|h\|_2^2} \leq K_\theta \leq \frac{M^{2L}}{16^L}, \quad \text{where:} \quad K_\theta = \frac{\prod_{k=1}^{L}\|\Theta_k\|_F^2}{16^L}. \tag{89}$$

$\square$

*Proof of Theorem 5.3.* More broadly, we have a natural constraint on the weights via the sigmoid activation function itself. As the weight values increase, the sigmoid function's derivative decreases, yielding progressively smaller weight updates. This diminishing effect results in the sigmoid function behaving like a constant function for large input values. Accordingly, there exists a numerical limit beyond which the sigmoid function output stabilizes at 1. This inherent characteristic of the sigmoid function suggests that an upper bound on $\|\Theta_k\|_F$ is given by the function itself, which removes the need for additional regularization constraints.

Following this observation, we now consider the following set:

$$A_0^j = \{\theta_j \mid \mathbb{P}\left[\phi\left(\theta_j^T y\right) > T\right] \geq \delta\}. \tag{90}$$

This set characterizes the parameters for which the sigmoid function reaches a certain threshold. Accordingly, larger values of the weights will not significantly contribute to the variation in the sigmoid function output. The aim is to find an upper bound on the possible values of $\theta_j$ (component-wise), such that $\theta_j \notin A_0^j$.

In a 64-bit floating point system, the machine epsilon, which defines the upper bound on relative error due to rounding in floating point arithmetic, is roughly $2.22 \times 10^{-16}$ (Higham, 2002). This is the precision limit, where two numbers closer than this difference can be considered indistinguishable at certain magnitudes. Since $y_i$ itself is the output of a sigmoid function, it takes values between $2.22 \times 10^{-16}$ and $1 - (2.22 \times 10^{-16})$, as values below and above these respective thresholds do not affect the numerical function output. Subsequently, to establish a suitable upper bound for $\theta_j$, we consider $y_i = 2.22 \times 10^{-16} \ \forall i \in \{1, \cdots, l\}$. We now require:

$$\mathbb{P}\left[\phi\left(\theta_j^T y\right) > T\right] = \mathbb{P}\left[\phi\left(\sum_{i=1}^{l} y_i \theta_{ji}\right) > T\right] = \mathbb{P}\left[\phi\left(2.22 \times 10^{-16} \sum_{i=1}^{l} \theta_{ji}\right) > T\right] \approx 1. \tag{91}$$

Let $\alpha = \sum_{i=1}^{l} y_i \theta_{ji}$. By definition, we know that $\alpha \geq \theta_{ji} \ \forall i \in \{1, \cdots, l\}$, so an upper bound on each $\theta_{ji}$ can be obtained by finding an appropriate $\alpha$. Thus, we can instead consider $\mathbb{P}[\phi(2.22 \times 10^{-16}\alpha) > T]$, as this is a stronger statement than that of Eq. (91). We must now choose $\alpha$, such that $\alpha > (\phi^{-1}(T))/(2.22 \times 10^{-16})$.

From the previous discussion, we recall that the maximum value of the threshold, $T$, that does not violate the precision limit is $T = 1 - (2.22 \times 10^{-16})$. We need $\alpha > (\phi^{-1}(1 - (2.22 \times 10^{-16})))/(2.22 \times 10^{-16}) \approx 1.6 \times 10^{17}$, so we choose $\alpha = 10^{18}$ to ensure that the inequality is satisfied. Altogether, we now have that:

$$\|g(y + h) - g(y)\|_2^2 \leq \frac{\|h\|_2^2}{16^L} \left(10^{36} \ d^2 \right)^L. \tag{92}$$

Therefore, as $K_\theta > 0$, we can consider the square of the Lipschitz constant to be bounded above by:

$$K_\theta^2 \leq \left(\frac{10^{36} \ d^2}{16}\right)^L \iff K_\theta \leq \sqrt{\left(\frac{10^{36} \ d^2}{16}\right)^L} = K. \tag{93}$$

$\square$

# D Manifold Generalizations

*Proof of Lemma 6.2.* Recall that $|U| = m$, and let $u_i$ denote the $i^{th}$ element of $U$. Leveraging the chain rule of differential entropy to expand $h(X_U)$, we obtain:

$$h(X_U) = h(X_{u_1}) + h(X_{u_2}|X_{u_1}) + \ldots + h(X_{u_m}|X_{u_1}, X_{u_2}, \ldots X_{u_{m-1}}). \tag{94}$$

From this Lemma, and given that conditioning cannot increase differential entropy, it follows that each term of Eq. (94) must be finite. As the sum of $m$ finite terms if finite, it follows that $h(X_U) > -\infty$. $\square$

**Remark D.1.** *While we do not provide a proof for the existence of a valid $U$ for any arbitrary data distribution, $\mathcal{D}$, one can use graph theory techniques to validate the existence of a valid $U$ for a specific choice of $\mathcal{D}$. Let $M \in \{0, 1\}^{d \times d}$ be a matrix such that $M_{i,j} = 1$ if $h(X_i|X_j) = -\infty$, and $M_{i,j} = 0$ if $h(X_i|X_j) > -\infty$. If $G$ is a directed graph with adjacency matrix $M$, then $U$ can be any set that is both an independent and absorbing set of $G$ (analogous to independent dominating sets in undirected graphs (Haynes et al., 2013)).*

*Proof of Theorem 6.3.* Recall that $x_U = (x_i)_{i \in U} \in \mathbb{R}^m, x_W = (x_i)_{i \in W} \in \mathbb{R}^{d-m}$. We analyze the autoencoder $_K\mathcal{A}_{d \mapsto l \mapsto d}$ via a six stage process. We suppose that the parameters of this autoencoder are fixed.

1. **Vector Element Manipulation:** Consider a function that extracts the elements of a vector, $x$, indexed by $U$: $\Pi_U(x) = x_U : \mathbb{R}^d \mapsto \mathbb{R}^m$. Similarly, let $\Pi_W(x) = x_W : \mathbb{R}^d \mapsto \mathbb{R}^{d-m}$. Finally, consider a function, $\Pi_*$, that interleaves the elements of vectors $x_U, x_W$: $\Pi_*(x_U, x_W) = x : \mathbb{R}^m \times \mathbb{R}^{d-m} \mapsto \mathbb{R}^d$.

2. **Transformations**: Consider $\Lambda^\dagger(x) : \mathbb{R}^m \mapsto \mathbb{R}^{d-m}$ as the vector of all $\Lambda_w : \Lambda^\dagger(x) = (\Lambda_w(x))_{w \in W}$. Suppose the function $\Sigma(x) : \mathbb{R}^m \mapsto \mathbb{R}^d = \Pi_*(x, \Lambda^\dagger(x))$. The function $\Sigma(x)$ concatenates $\Lambda^\dagger(x)$ with $x$ to form a $\mathbb{R}^d$ column vector. For any vector, $x$, $\Pi_U(\cdot) = \Sigma^{-1}(\cdot)$, and hence $\Sigma(\Pi_U(x)) = x$.

3. **Preprocessing:** We first define the function:

$$\Gamma_{m,d}(\cdot) \triangleq \ _K\mathcal{A}_{d \mapsto l \mapsto d}(\Sigma(\cdot)) : \mathbb{R}^m \mapsto \mathbb{R}^d. \tag{95}$$

   We note that for any $x \sim \mathcal{D}$, where $x_U = \Pi_U(x)$:

$$\Gamma_{m,d}(x_U) \triangleq \ _K\mathcal{A}_{d \mapsto l \mapsto d}(x). \tag{96}$$

4. **Postprocessing:** We now define the function:

$$\Gamma_{m,m}(\cdot) \triangleq \Pi_U(\Gamma_{m,d}(\cdot)) : \mathbb{R}^m \mapsto \mathbb{R}^m. \tag{97}$$

   Furthermore, we define $\hat{x} = \ _K\mathcal{A}_{d \mapsto l \mapsto d}(x)$. It now follows that:

$$\|\Gamma_{m,m}(\Pi_U(x)) - \Pi_U(x)\|^2 = \sum_{u \in U}(\hat{x}_i - x_i)^2,$$

   and:

$$\| \ _K\mathcal{A}_{d \mapsto l \mapsto d}(x) - x\|^2 = \sum_{i=1}^d (\hat{x}_i - x_i)^2.$$

   Moreover, as $U \subseteq \{1, 2, \ldots, d\}$, it follows that:

$$\sum_{u \in U}(\hat{x}_i - x_i)^2 \le \sum_{i=1}^d (\hat{x}_i - x_i)^2.$$

   Therefore:

$$\|\Gamma_{m,m}(\Pi_U(x)) - \Pi_U(x)\|^2 \le \| \ _K\mathcal{A}_{d \mapsto l \mapsto d}(x) - x\|^2. \tag{98}$$

5. **Bound:** We observe that all the components of $\Gamma_{m,m}$ are deterministic for fixed input, $x$. Moreover, the output of $\Gamma_{m,m}(x)$ is in $[0,1]^m$. Define the decoder of $_K\mathcal{A}_{d \mapsto l \mapsto d}(x)$ as $\mathbb{D}_{l \mapsto d}(x) : \mathbb{R}^l \mapsto \mathbb{R}^d$. If $\mathbb{D}_{l \mapsto d}(x)$ is $K$-Lipschitz, then the decoder of $\Gamma_{m,m}$ is $\Pi_U(\mathbb{D}_{l \mapsto d}(x)) : \mathbb{R}^l \mapsto \mathbb{R}^m$, and is $K$-Lipschitz.

   Altogether, we observe that $\Gamma_{m,m}$ defines a $m \mapsto l \mapsto m$ autoencoder, which reconstructs $x_U$ into $\hat{x}_U$ after compressing it through a latent layer of dimension $l$. Since $\Gamma_{m,m}$ satisfies all conditions of Theorem 3.5, we can lower bound the generalization MSE of $\Gamma_{m,m}$ via the bound of Theorem 3.5:

$$\mathbb{E}[\|\Gamma_{m,m}(\Pi_U(X)) - \Pi_U(X)\|^2] \ge \mathcal{F}^*_{(m,l,h(X_U),K)}. \tag{99}$$

6. **Conclusion:** Combining Eq. (98) and Eq. (99), and substituting $|U| = m$, we finally arrive at:

$$\mathbb{E}[\| \ _K\mathcal{A}_{d \mapsto l \mapsto d}(X) - X\|^2] \ge \mathcal{F}^*_{(|U|,l,h(X_U),K)}. \tag{100}$$

The inequality of Eq. (100) must hold for any $U$ satisfying Assumption 6.1. This $U$ is not necessarily unique. Recall that $\mathbb{U}$ is the collection of all sets, $U$, that satisfy the definition in Assumption 6.1. Correspondingly, it follows that:

$$\mathbb{E}[\| \ _K\mathcal{A}_{d \mapsto l \mapsto d}(X) - X\|^2] \ge \max_{U \in \mathbb{U}} \mathcal{F}^*_{(|U|,l,h(X_U),K)}. \tag{101}$$

$\square$

*Proof of Corollary 6.4.* Let $X = (X_1, X_2, \ldots, X_d)$ be a sample of $\mathcal{D}$. Denote the marginal expectations as $\mathbb{E}[X_i]$. Then, this theorem is a special case of Theorem 6.3, where for all $i \in W$, $\Lambda_i(\cdot) = \mathbb{E}[X_i]$. $\square$

# E    Empirical Results Discussion

In this section, we provide descriptions of the datasets utilized to generate the empirical results, and outline the neural network architectures and hyperparameter choices. The empirical results were compiled using an NVIDIA GeForce RTX 3090 GPU.

## E.1    Dataset Descriptions

### E.1.1    Uniform Distribution Dataset

The uniform distribution dataset, $\mathcal{U}$, is a collection of samples derived from a $d$-dimensional uniform distribution, dennoted as $\mathcal{U}([0, 1]^d)$. For the reconstruction and regression examples from Section 7.1 of the main text, we consider $|\mathcal{I}| = 50$ training samples, and for the neural architecture search example from Section 7.2 of the main text, we consider $|\mathcal{I}| = 5000$ training samples. We consider $|\mathcal{I}_{\text{val}}| = 10{,}000$ test samples in both cases. The labels for the regression task are determined by first applying a linear transformation comprising a $d \times k$-dimensional standard normally distributed weight matrix and a $k$-dimensional standard normally distributed bias vector, and then adding $k$-dimensional Gaussian noise with standard deviation 0.1. The features and labels within the uniform distribution dataset are summarized as follows:

- Each feature, $x_i$, from the feature vector, $x$, is sampled from a Uniform, $\mathcal{U}(0, 1)$.

- Target Variable: The $k = 1$-dimensional labels from a linear transformation on the feature vector.

### E.1.2    Truncated Normal Distribution Dataset

The truncated normal distribution dataset, $\mathcal{TN}$, is a collection of samples derived from a $d$-dimensional truncated normal distribution, $\mathcal{TN}([0, 1]^d)$. For the reconstruction and regression examples from Section 7.1 of the main text, we consider $|\mathcal{I}| = 50$ training samples, and for the neural architecture search example from Section 7.2 of the main text, we consider $|\mathcal{I}| = 5000$ training samples. We consider $|\mathcal{I}_{\text{val}}| = 10{,}000$ test samples in both cases. The labels for the regression task are formed by first applying a linear transformation comprising a $d \times k$-dimensional standard normally distributed weight matrix and a $k$-dimensional standard normally distributed bias vector, and then adding $k$-dimensional Gaussian noise with standard deviation 0.1. The features and labels within the truncated normal distribution dataset are summarized as follows:

- Each feature, $x_i$, from the feature vector, $x$, is sampled from a Truncated Normal, $\mathcal{TN}(0, 1, 0, 1)$.

- Target Variable: The $k = 1$-dimensional labels from a linear transformation on the feature vector.

### E.1.3    MNIST Dataset

The **MNIST** (Modified National Institute of Standards and Technology) dataset is a collection of handwritten digits commonly used to train image processing systems. For the reconstruction example from Section 7.1 of the main text, we consider $|\mathcal{I}| = 500$ training samples, and for the classification example from Section 7.1 of the main text, we consider $|\mathcal{I}| = 20$ training samples. We consider $|\mathcal{I}_{\text{val}}| = 10{,}000$ test samples in both cases. The features and labels within the MNIST dataset are summarized as follows:

- Each feature vector, $x$, is of size 784 ($28 \times 28$), and has normalized grayscale intensities from 0 to 1.

- Target Variable: The numerical class (digit) the image represents, ranging from 1 to $k = 10$.

### E.1.4 Neural Network Architectures

We now provide detailed descriptions of several different neural network architectures designed for reconstruction, and regression/classification with implicit reconstruction. Each of these architectures were employed to generate the respective empirical results pertaining to the aforementioned tasks.

### E.1.5 Reconstruction

We utilize autoencoders in our analysis for reconstruction tasks. For reconstruction on the uniform distribution, truncated normal, and MNIST datasets, the network consists of the following layers, where $L = 2$:

- **Fully Connected Layer (fc1)**: Transforms the $d$-dimensional input into a $(d - \frac{d-l}{2})$-dimensional space.

- **Sigmoid Activation (sigmoid1)**: Applies the Sigmoid activation function to the output of fc1.

- **Fully Connected Layer (fc2)**: Takes the sigmoid1 output and maps it to a $l$-dimensional space.

- **Sigmoid Activation (sigmoid2)**: Applies the Sigmoid activation function to the output of fc2.

- **Fully Connected Layer (fc3)**: Expands the $l$-dimensional latent space to a $(l + \frac{d-l}{2})$-dimensional space.

- **Sigmoid Activation (sigmoid3)**: Applies the Sigmoid activation function to the output of fc3.

- **Fully Connected Layer (fc4)**: Takes the sigmoid3 output and maps it to a $d$-dimensional space.

- **Sigmoid Activation (sigmoid4)**: Applies the Sigmoid activation function to the output of fc4.

### E.1.6 Regression/Classification with Implicit Reconstruction

For the regression/classification test cases with implicit reconstruction, we leverage fully connected networks for regression tasks on the uniform distribution and the truncated normal distribution datasets, and for classification tasks on MNIST. The network consists of the following layers:

- **Fully Connected Layer (fc1)**: Transforms the $d$-dimensional input into a $(d - \frac{d-l}{2})$-dimensional space.

- **Sigmoid Activation (sigmoid1)**: Applies the Sigmoid activation function to the output of fc1.

- **Fully Connected Layer (fc2)**: Takes the sigmoid1 output and maps it to a $l$-dimensional space.

- **Sigmoid Activation (sigmoid2)**: Applies the Sigmoid activation function to the output of fc2.

- **Fully Connected Layer (fc3)**: Expands the $l$-dimensional latent space to a $(l + \frac{d-l}{2})$-dimensional space.

- **Sigmoid Activation (sigmoid3)**: Applies the Sigmoid activation function to the output of fc3.

- **Fully Connected Layer (fc4)**: Takes the sigmoid3 output and maps it to a $k$-dimensional output space.

We train this architecture on the regression/classification datasets from Section E.1 of the Appendix. To leverage implicit reconstruction, we append a separate decoder to the latent space, which accepts detached $l$-dimensional latent representations as inputs. This network consists of the following layers:

- **Fully Connected Layer (fc3)**: Expands the $l$-dimensional latent space to a $(l + \frac{d-l}{2})$-dimensional space.

- **Sigmoid Activation (sigmoid3)**: Applies the Sigmoid activation function to the output of fc3.

- **Fully Connected Layer (fc4)**: Takes the sigmoid3 output and maps it to a $d$-dimensional space.

- **Sigmoid Activation (sigmoid4)**: Applies the Sigmoid activation function to the output of fc4.

## E.2 Neural Network Training Hyperparameters

The relevant hyperparameters used to train the neural networks from Appendix Section E.1.4 are provided in Table 2. All results presented in the main text were produced using these hyperparameter choices. We note that the learning rates for the regression and classification (with implicit reconstruction) experiments pertain to the fully connected networks, wherein the appended decoders have learning rate $\alpha = 0.005$.

Table 2: Neural Network Training Hyperparameters (grouped by experiment).

| Dataset | Experiment | Optimizer | Learning Rate ($\alpha$) | Training Dataset Size ($N$) |
|---|---|---|---|---|
| MNIST | Reconstruction | Adam | 0.005 | 500 |
| $\mathcal{U}$ | Reconstruction | Adam | 0.005 | 20 |
| $\mathcal{TN}$ | Reconstruction | Adam | 0.005 | 20 |
| MNIST | Classification | Adam | 0.005 | 20 |
| $\mathcal{U}$ | Regression | Adam | 0.005 | 50 |
| $\mathcal{TN}$ | Regression | Adam | 0.005 | 50 |
| $\mathcal{U}$ | Neural Architecture Search | Adam | 0.005 | 5000 |
| $\mathcal{TN}$ | Neural Architecture Search | Adam | 0.005 | 5000 |

## F Lossless Compression

To motivate our proposed lower bound, we examine the impossibility of having *continuous* bijective maps to perform compressive transformations when $d > l$. However, one can still construct a *discontinuous* bijection between $\mathbb{R}^d$ and $\mathbb{R}^l$ for $d > l$. A standard example is the interleaving construction (see Lemma F.1).

**Lemma F.1** (Interleaving Bijection). *For all positive integers $d, l \in \mathbb{Z}^+$, with $d > l$, there exists a bijection:*

$$f : \mathbb{R}^d \longrightarrow \mathbb{R}^l. \tag{102}$$

*Proof.* Since $\mathbb{R}$ is in bijection with the open interval $(0, 1)$ (for example via $x \mapsto \frac{1}{\pi} \arctan(x) + \frac{1}{2}$), it suffices to construct a bijection between $(0,1)^d$ and $(0,1)^l$, which is based on Cantor's 1877 argument (Malek et al., 2010). We do so by interleaving and "de-interleaving" decimal digits. We first consider:

$$x = (x_1, \ldots, x_d) \in (0, 1)^d, \tag{103}$$

and write each coordinate in its (non-terminating-with-all-9's) decimal expansion:

$$x_i = 0. \, x_{i,1} x_{i,2} x_{i,3} \cdots, \quad \text{where:} \quad x_{i,j} \in \{0, \ldots, 9\}, \quad \text{and:} \quad \exists i, j, \ x_{i,j} \neq 0 \vee x_{i,j} \neq 9 \tag{104}$$

Subsequently, we now define:

$$f(x) \;=\; 0. \, x_{1,1} \, x_{2,1} \, \ldots \, x_{d,1} \, x_{1,2} \, \ldots \, x_{d,2} \, x_{1,3} \, \ldots \;=\; \sum_{j=1}^{\infty} \sum_{i=1}^{d} \frac{x_{i,j}}{10^{\, i + d\,(j-1)}} \in (0, 1), \tag{105}$$

This map $\Psi \colon (0,1)^d \to (0,1)$ is bijective under the rule that no expansion ends in an infinite tail of 9's. Next, we consider that for any:

$$y = 0. \, y_1 y_2 y_3 \cdots \in (0, 1), \tag{106}$$

we can "de-interleave" to obtain $\Xi(y) \in (0, 1)^l$ by grouping digits in blocks of length $l$. Specifically, for each $i \in \{1, \ldots, l\}$, we have that:

$$\big(\Xi(y)\big)_i = 0. \, y_i \, y_{i+l} \, y_{i+2l} \cdots \;=\; \sum_{j=1}^{\infty} \frac{y_{\, i+(j-1)l}}{10^j} \in (0, 1). \tag{107}$$

Again forbidding infinite 9-tails ensures $\Xi$ is a bijection from $(0,1)$ onto $(0,1)^l$. Finally, composing bijections:

$$\mathbb{R}^d \xrightarrow{\cong} (0,1)^d \xrightarrow{\Psi} (0,1) \xrightarrow{\Xi} (0,1)^l \xrightarrow{\cong} \mathbb{R}^l. \tag{108}$$

This yields the desired bijection, $f : \mathbb{R}^d \to \mathbb{R}^l$. $\qquad\qquad\square$

Lemma F.1 demonstrates that we *could* compress the continuous-valued input image into a lower-dimensional continuous-valued latent representation with zero loss and zero reconstruction MSE. Even so, the encoder of our MLP could never learn the function of Lemma F.1, since this function is *discontinuous*. The MLPs we consider in this paper are compositions of multiplication by weights, addition with biases, and transformation by sigmoid activations, each of which are continuous, and hence the MLP is continuous. Moreover, the result of Cybenko holds only for continuous target functions (Cybenko, 1989). In fact, it can be shown that there does *not* exist any continuous function, $f$, which bijects between $\mathbb{R}^d$ and $\mathbb{R}^l$:

**Lemma F.2** (No Continuous Bijection Exists Between $\mathbb{R}^d$ and $\mathbb{R}^l$ when $d > l$). *For all positive integers $d, l \in \mathbb{Z}^+$, with $d > l$, there does not exist a continuous bijection:*

$$f : \mathbb{R}^d \longrightarrow \mathbb{R}^l. \tag{109}$$

*Proof.* Assume, for the sake of contradiction, that such a map $f$ exists. Because $f$ is bijective, it is in particular injective. Consider a point $p \in \mathbb{R}^d$ and let $U$ be an open ball around $p$; the restriction $f|_U : U \to \mathbb{R}^l$ remains continuous and injective. Brouwer's Invariance-of-Domain Theorem asserts that a continuous injective map between Euclidean spaces of the same dimension sends open sets to open sets and is a homeomorphism onto its image (Brouwer, 1912). Applying the theorem to $f|_U$, we obtain that $f(U)$ is open in $\mathbb{R}^l$ and that:

$$f|_U : U \longrightarrow \Xi(U), \tag{110}$$

is a homeomorphism. Consequently $U$ and $f(U)$ are homeomorphic manifolds. The topological invariance of dimension, an immediate corollary of the same theorem, states that Euclidean manifolds are homeomorphic only when their dimensions coincide. Hence:

$$\dim U = d = \dim f(U) = l, \tag{111}$$

contradicting the hypothesis $d > l$. Therefore no continuous bijection $f : \mathbb{R}^d \to \mathbb{R}^l$ can exist when $d > l$. $\quad\square$

We now discuss a consequence of Lemma F.2. Define $d = l = 1^\dagger$. Suppose we let $g(y) = Ky$ and $f(x) = x/K$ for some large value of $K$. Then, $g(f(x)) = x$ and the autoencoder has zero MSE. As the range of the latent layer is $[0,1]$, a very small $K$ implies that the distribution over images $x$ is contained in some $[a,b]$ such that $|b-a|\frac{1}{K} \leq 1$. Hence, for $K > 1$, the maximum entropy distribution over $x$ is the uniform distribution over $[a,b] \subseteq [0,1]$ and this distribution has differential entropy $\log|b-a|$. We recall that:

$$r(s) \triangleq \left(\frac{s}{s+1/4}\right)^{-\frac{l}{d}} \sqrt{s}, \tag{112}$$

and that:

$$\mathcal{F}_\eta(s) > 0 \iff \frac{\alpha\beta}{\gamma} > r(s), \tag{113}$$

by Lemma B.1. In this setting, we observe that:

$$\alpha = \frac{1}{2\pi e}, \quad \beta = \exp(2h) = \exp(\log|b-a|2) = (b-a)^2, \quad \gamma = 2K. \tag{114}$$

Plugging $l = d = 1$ into $r(s)$, we observe that $r(s)$ can be expressed as:

$$r(s) = \left(\frac{s+1/4}{s}\right)\sqrt{s}, \tag{115}$$

---

$^\dagger$We note that the discussion of this section neither follows from nor contradicts the results of Theorem 4.5 as it is conditioned upon Assumption 4.4, which does not hold for all $K$ in this example.

and that this function attains a global minimum of one. We further observe that:

$$\frac{\alpha\beta}{\gamma} = \frac{(b-a)^2}{2\pi e 2K} = \frac{(K)^2}{2\pi e 2K} = \frac{K}{4\pi e}. \tag{116}$$

Therefore, as $K$ gets small, $\alpha\beta/\gamma$ approaches zero. We observe that for all $K < 1$, that $\alpha\beta/\gamma < 1$. Thus, for all $K < 1$, we know that $\mathcal{F}_\eta^* = 0$.

