# OpenReview forum: "An Information-Theoretic Lower Bound on the Generalization Error of Autoencoders"
_TMLR — Accepted by TMLR_

### Review · Reviewer_5oFc · 2025-06-29

**Summary Of Contributions:**

Authors give an information-theoretic lower bound on the generalization mean squared error of autoencoders with sigmoid activation functions using the Estimation Error and Differential Entropy (EEDE) inequality for continuous random vectors
The MSE loss of an autoencoder $\theta$ with $\mathcal{A}_\theta(x) = g(f(x))$ where $f: \mathbb{R}^d \rightarrow \mathbb{R}^l$ and $g: \mathbb{R}^l \rightarrow \mathbb{R}^d$ is given by

$$  \frac{1}{n}  \sum_{i=1}^n \Vert x_i - \mathcal{A}_\theta(x)  \Vert_2^2.$$
The main result of the paper is to approximate the MSE by $\mathcal{F}$ which depends on the varaiance of an induced injective noise varaible, as well as the dimensions $d$ and $l$, the entropy and $K$ is the supremum of the Lipschitz constants of the auto encoders.
The bound is provided under different assumptions and is strictly greater than $0$ if $l <d/2$.
Experiments with respect to overfitting detection in reconstruction, regression, and classification tasks are provided as well.

**Audience:**

Yes

**Claims And Evidence:**

Yes

**Requested Changes:**

Can you explain why the conditional entropy h(Y |X) is $-\infty$ if Y=f(X)?

**Strengths And Weaknesses:**

The write up of the paper is well structured and clear.
The problem considered in the paper is interesting.
The proofs seem to be be fairly straight forward but non trivial.
Experiments support the theoretical claims.

---

> ### Author Response · Authors · 2025-07-14
> **Rebuttal by Authors**
>
> Dear Reviewer 5oFc,
>
> Thank you for your helpful feedback. Below, we have addressed your comments.
>
> > Can you explain why the conditional entropy $h(Y |X)$ is $-\infty$ if $Y=f(X)$?
>
> The amount of uncertainty one has regarding the observations of a random variable is quantified by the differential entropy, $h(x) = \int_{\mathbb{R}^d} -p(x) \log p(x) dx$. If $y=f(x)$, then once one knows $x$, $y$ is determinstically known; there is no uncertainty, and the conditional differential entropy is negative infinity.
>
> Thank you again for your positive feedback.

---

> ### Comment · Reviewer_5oFc · 2025-07-20
>
> This should be $0 \cdot \infty$ and in the case of entropy this should be $0$ as $\lim_{p\rightarrow 0}p\log(p)=0$. At least this is what I'm familar with.

---

> > ### Author Response · Authors · 2025-07-22
> > **Rebuttal by Authors**
> >
> > Thank you for your follow‑up. We agree with the reviewer that the conditional _discrete_ entropy in this degenerate case is $0$. However, the corresponding conditional _differential_ entropy is $-\infty$.
> >
> > A Gaussian limiting argument makes this explicit.  Let $Z_\sigma \sim \mathcal{N}(0,\sigma^2)$, so that $h(Z_\sigma) = \frac{1}{2} \ln(2\pi e \sigma^2) \longrightarrow -\infty$ as $\sigma \to 0$.
> > The random variable $Y|X$ when $Y = f(X)$ follows a density that is a dirac delta function. The dirac delta function is the limit of $\mathcal{N}(0, \sigma^2)$ as $\sigma \rightarrow 0$.  Hence, $h(Y|X)$ is equivalent to $\lim_{\sigma \rightarrow 0} h(Z_\sigma) = -\infty$.
> >
> >
> > Cover and Thomas [1] further state that whenever a continuous random variable collapses onto a lower‑dimensional manifold (here, the graph of $Y=f(X)$), its differential entropy is $-\infty$ (see p. 249 in the link provided below).
> >
> > **References:**
> >
> > [1] T.M. Cover and J.A. Thomas, _Elements of Information Theory_, 2nd ed., Wiley, 2006. https://dl.icdst.org/pdfs/files/aea72e61329cd4684709fa24f15ac098.pdf

---

> > > ### Comment · Reviewer_5oFc · 2025-07-22
> > >
> > > I get it now, thanks for the clarification!

---

### Review · Reviewer_5daf · 2025-07-04

**Summary Of Contributions:**

This paper provides a theoretical analysis of the generalization error of deep autoencoder (AE) neural networks, in which they establish a lower bound on the generalization error. This bound depends on the dimension of the AE embedding, the smoothness of the decoder, and the complexity of the data, and the bound is derived from the inability of a smooth decoder being able to reconstruct complex high-dimensional data from a lower-dimensional representation. The authors provide practical guidance for how to bound or estimate each of the required quantities to obtain a true bound on the generalization mean squared error under reasonable assumptions. In experiments with simulation data and a subset of MNIST images, the authors demonstrate how the bound can be used to detect overfitting (without a validation set) or guide architecture optimization (by eliminating architectures that are theoretically incapable of achieving a small error).

**Audience:**

Yes

**Claims And Evidence:**

Yes

**Requested Changes:**

- In the final line of Section 3.1, s* is referenced before it is defined later in Section 4.

**Strengths And Weaknesses:**

This is a well-written paper describing high-quality work. A very strong contribution with many strengths:
- The presentation is excellent. The logic is clear and easy to follow. It was a joy to read!
- While there is tremendous interest in theoretical analysis of deep neural network architectures, it has proven very difficult to obtain theoretical bounds with any practical value. In this work, the authors complete the connection from theoretical bound to decisions. Even though the demonstrations were only on tiny datasets, I found them compelling and impressive.
- The experiments demonstrating transfer learning to classification tasks are a nice addition.
- Even though the proposed bound may not be informative for most practical applications, the results and proposed techniques will be of interest to the community.

Weaknesses:
- Ultimately, the bounds are only useful for toy problems. In most applications of neural networks, practitioners are using larger models for which this bound would not be informative.

---

> ### Author Response · Authors · 2025-07-14
> **Rebuttal by Authors**
>
> Dear Reviewer 5daf,
>
> Thank you for your positive feedback. Below, we have provided responses to your comments.
>
> > Ultimately, the bounds are only useful for toy problems. In most applications of neural networks, practitioners are using larger models for which this bound would not be informative.
>
> We agree that the numerical simulations provided are of moderate dimension and (for the datasets of analytical distributions) toy in nature. The main contribution of this work is theoretical, and the simulations are provided principally to illustrate the bound's properties.
>
> Extending and generalizing this theory remains an important part of our ongoing research.
>
> > In the final line of Section 3.1, s* is referenced before it is defined later in Section 4.
>
> We thank the reviewer for catching this and have corrected it in our revision.
>
> Thank you again for your encouraging feedback.

---

### Review · Reviewer_caDN · 2025-07-11

**Summary Of Contributions:**

This paper presents an information-theoretic lower bound on the generalization error (as opposed to the generalization gap) of autoencoders with sigmoid activation functions. The authors derive this bound using differential entropy and investigate how various parameters influence it. Notably, the derivation relies on minimal assumptions regarding the data distribution and the autoencoder architecture. Finally, the paper provides empirical results illustrating the application of this bound to overfitting detection and neural architecture search.

**Audience:**

Yes

**Broader Impact Concerns:**

None from my end.

**Claims And Evidence:**

Yes

**Requested Changes:**

1. **[Critical]** For practical use of this bound I still have 1 major question. The bound is on the generalization MSE (i.e., loss over the distribution) and not on the training dataset $\mathcal{I}$, if one wants to relate this to the training loss then one would need to use some concentration inequality to bound the exact training loss with probability. However, in section 7 for overfitting and neural architecture search authors directly use the bound for entire distribution and compare it with training loss, can the authors comment on it?
2. **[Critical]** The bound does not depend on the structure of encoder(f) and decoder(g). Can they be attention modules? Is it necessary for f to be a neural network? Architectures do create significant difference in overfitting e.g., for image CNN works much better than MLP as they tend to overfit also adding dropout reduces the overfitting, so even if I do any such change in architecture the lower bound for detection of  overfitting remains same. This raise question about the tightness of the bound, is the bound tight for MLPs or Deep neural networks?
3. **[Critical]** The lower bound has no dependence on the weights of the neural networks. Only decoder has a Lipschitz constraint but this constant is loosely upper bounded and does not significantly impact the error bound. Even for neural architecture search only latent dimension could be searched. This limits the use of bound for future usage as it’s invariant to all other parts. This is the major drawback.
4. **[Critical]** The most informative bound assumes K = 10^5, however for just 2-3 layer NN the **actual Lipschitz constant** could be very small because of the normalization layers in decoders and normalized output of encoder, will the assumption of s << 1/4 still hold? Moreover, if it’s only valid for more deeper layers then the problem of incorrect estimation of differential entropy can come in (please correct me if I am wrong). I am unsure if the bound will remain practical for real value of K. Have the authors tried calculating the actual value of K (or using some tighter bound) in the experiments?
5. If the authors could highlight the Lipschitz constant is over the input of decoder in the main paper (instead of keeping this in the appendix) it will be very helpful for the readers. In generalization literature people often use Lipschitz constant w.r.t the model parameters which could cause confusion.
6. In Assumption 6.1 equation 22, the constraint still can’t handle some types of features, like when $x_i = 2* x_j$ and $Var(x_i) ≠ 0$. This could just be highlighted in limitations.
7. In Section 7.1.2 where authors use the bound for regression and classification task, the hypothesis is not backed up by any theory and it blames the encoder (f) for overfitting. I am not convinced with this argument, and the approach of extending current bounds (on reconstruction) for classification and regression task.

**Strengths And Weaknesses:**

My overview of the paper is positive, from the application of Differential entropy to estimating the parameters of bound all seems reasonable and properly justified. However, I still have some concerns about the paper which I have raised in **Requested Changes** section.

# Strengths

- The motivation of the paper is very clear to get useful lower bound on the generalization error, they derive the lower bound and also show it’s usefulness in practical setting. The paper is well written with proper reasons, theorems and proofs.
- Although the use of noise to overcome trivial bounds is popular in information-theoretic domain this approach according to me has not been used to study the generalization error of autoencoders.
- Based on Theorem 3.5 the terms on R.H.S form correct intuition as highlighted in paragraph after Lemma 4.2. An increase in the Lipschitz constant of decoder shows a decrease in lower bound which fits correctly. This is often not observed in literature of generalization gap bounds, where an upper bound on the Lipschitz constant (over parameters of NN) is required to show generalization.
- The assumption 4.3 and 4.4 seems practical and assumption 6.1 extends it's scope further.
- The estimation of bound parameters was much needed effort which authors did to make the theorem practically applicable. The bounds although not tight but have clear explanation of why a tighter bound is not required and it is easily usable.

# Weaknesses

- I have mainly 2-3 points as weakness. I have mentioned them in Requested Change, they are mostly regarding the usage of this bound, tightness of the bound and non-dependence of the bound on other quantities of neural network.
- One which are critical to secure my recommendation for acceptance are tagged with [Critical] at start.

---

> ### Author Response · Authors · 2025-07-14
> **Rebuttal by Authors**
>
> Dear Reviewer caDN,
>
> Thank you for your valuable feedback. Below, we have addressed your comments.
>
> > For practical use of this bound I still have 1 major question. The bound is on the generalization MSE (i.e., loss over the distribution) and not on the training dataset, if one wants to relate this to the training loss then one would need to use some concentration inequality to bound the exact training loss with probability. However, in section 7 for overfitting and neural architecture search authors directly use the bound for entire distribution and compare it with training loss, can the authors comment on it?
>
> Though the training data points are drawn i.i.d. from a distribution $\mathcal{D}$ (and hence $\mathcal{D}^{\otimes N}$ is a probability measure over the set of all possible training datasets of length $N$), the training dataset $\mathcal{I}$ is a particular realization of this distribution.  Once training begins, all optimization and learning is conditioned upon one particular observation, $\mathcal{I}$.  Hence, the training loss (e.g., empirical risk) is not a sample mean estimator of the moment of some distribution (unlike the generalization loss / population risk); rather, the training loss is _defined to be_ a sample mean evaluated over the images in the training dataset.  As such, applying a concentration inequality to the training loss is not necessary (and in our case is not even well-defined).
>
> Even so, in some of our results, including the results of Figure 2 and 3, we provide an estimate of uncertainty in our plots (determined by repeating a training process multiple times, though each training process uses the same dataset $\mathcal{I}$).
>
> > The bound does not depend on the structure of encoder(f) and decoder(g). Can they be attention modules? Is it necessary for f to be a neural network? Architectures do create significant difference in overfitting e.g., for image CNN works much better than MLP as they tend to overfit also adding dropout reduces the overfitting, so even if I do any such change in architecture the lower bound for detection of overfitting remains same. This raise question about the tightness of the bound, is the bound tight for MLPs or Deep neural networks?
>
> Yes, our lower bound supports attention and convolution layers in the autoencoder provided that:
> - Each layer is continuous.
> - The codomain of the encoder, $f(\cdot)$, is $(0,1)^l$.
>
> We note that our method of estimating the Lipschitz constant of the decoder, $g(\cdot)$, imposes further assumptions which may or may not hold for these layer types. However, the bound is valid for the constraints provided.  We do not claim tightness (see Section 8), and echo the reviewer's previous comment that the paper explains why _a tighter bound is not required and it is easily usable_.
>
> > The lower bound has no dependence on the weights of the neural networks. Only decoder has a Lipschitz constraint but this constant is loosely upper bounded and does not significantly impact the error bound. Even for neural architecture search only latent dimension could be searched. This limits the use of bound for future usage as it’s invariant to all other parts. This is the major drawback.
>
> We provide one illustration involving neural architecture search to demonstrate the utility of the proposed bound, but we emphasize that our primary contribution is theoretical. We provide four hyperparameters for our bound: $\eta = (d, l, K, h_\mathcal{D})$, as per Equation 10.  We believe that a researcher who wishes to utilize the theory of our bound for neural architecture search has a fixed dataset, $\mathcal{I}$, of images from a distribution, $\mathcal{D}$.  The input dimension of the autoencoder, $d$, and the differential entropy of the data distribution, $h_\mathcal{D}$, are _entirely determined by the data distribution_, $\mathcal{D}$, which we fix and do not optimize. We do not perform optimization over the decoder Lipschitz constant as the dependence between it and the bound is negligible for large $K$ (Theorem 5.1).
>
> We emphasize that our neural architecture search method is able only to exclude small latent dimensions from consideration and that it cannot directly calculate the optimal choice of $l$.
>
> (response continued in comment)

---

> ### Author Response · Authors · 2025-07-14
> **Continuation of Rebuttal**
>
> > The most informative bound assumes K = $10^5$, however for just 2-3 layer NN the actual Lipschitz constant could be very small because of the normalization layers in decoders and normalized output of encoder, will the assumption of $s << 1/4$ still hold? Moreover, if it’s only valid for more deeper layers then the problem of incorrect estimation of differential entropy can come in (please correct me if I am wrong). I am unsure if the bound will remain practical for real value of K. Have the authors tried calculating the actual value of K (or using some tighter bound) in the experiments?
>
> We note that Figure 1 (which considers $K = 10^{5}$) is illustrative only.  The parameters chosen are presented solely to illustrate the shape and qualitative behavior (in the practical regime) of the bound; these parameters do not correspond to any of the distributions simulated in the empirical results from Section 7.  Furthermore, the assumption that $s^* << \frac{1}{4}$ merely enables a numerical approximation that can simplify the identification of the maximum value achieved by the bound.  As $\hat{\mathcal{F}}\_{\eta}^* \leq \mathcal{F}\_{\eta}^*$ under this approximation, our proposed bound is not violated.
>
> The differential entropy, $h_\mathcal{D}$, depends only upon the data distribution, $\mathcal{D}$. The estimation of differential entropy does not have any relationship or dependence upon the neural network, including the Lipschitz constant, $K$, of the decoder, $g(\cdot)$.
>
> We further note that even for a 2-3 layer neural network, the Lipschitz constant can still be extremely large. Suppose that $l = 1, d=10$ and that $g(0.5) = 0_d, g(0.5 + 10^{-7}) = 1_d$. This construction permits a larger Lipschitz constant.
>
> > If the authors could highlight the Lipschitz constant is over the input of decoder in the main paper (instead of keeping this in the appendix) it will be very helpful for the readers. In generalization literature people often use Lipschitz constant w.r.t the model parameters which could cause confusion.
>
> We thank the reviewer for their feedback and have updated the main text to incorporate this helpful suggestion.
>
> > In Assumption 6.1 equation 22, the constraint still can’t handle some types of features, like when $x_i = 2 \times x_j$ and $\text{Var}(x_i) > 0$. This could just be highlighted in limitations.
>
> We note that if $x_i = 2 \times x_j$ with $\text{Var}(x_i) > 0$, then there exists a function $\Lambda$ such that $\Lambda(x_i) = x_j$, hence one of $x_i$ or $x_j$ must be in the set $U$ of Assumption 6.1.  As such, this case is explicitly handled by Theorem 6.3 and is not a limitation.
>
> > In Section 7.1.2 where authors use the bound for regression and classification task, the hypothesis is not backed up by any theory and it blames the encoder (f) for overfitting. I am not convinced with this argument, and the approach of extending current bounds (on reconstruction) for classification and regression task.
>
> The result of Section 7.1.2 is a study presented to depict a trend observed regarding the overfitting of classification and regression networks in relation to autoencoders.  We emphasize that the main contribution of this paper is theoretical.
>
> We believe that the presentation of this result appropriately reads as argument rather than theoretical proof: in Section 7, we state that _we hypothesize that the presence of overfitting in this corresponding autoencoder implies overfitting in $t(f(\cdot))$_, and in the _Motivation_ paragraph of Section 1, we state that _we further hypothesize that this bound can be leveraged to suggest overfitting in regression and classification tasks_.
>
> Thank you again for your feedback, which has been instrumental in improving our paper.

---

> > ### Comment · Reviewer_caDN · 2025-07-21
> > **Rebuttal Reply**
> >
> > I thank the authors for a thorough reply. I am satisfied with most of the answers to my questions. Since, we have some time I would like to clear up few of my minor concerns.
> >
> > > Though the training data points are drawn i.i.d. from a distribution $\mathcal{D}$ (and hence is a probability measure over the set of all possible training datasets of length $\mathcal{N}$), the training dataset $\mathcal{I}$ is a particular realization of this distribution. Once training begins, all optimization and learning is conditioned upon one particular observation, $\mathcal{I}$. Hence, the training loss (e.g., empirical risk) is not a sample mean estimator of the moment of some distribution (unlike the generalization loss / population risk); rather, the training loss is defined to be a sample mean evaluated over the images in the training dataset. As such, applying a concentration inequality to the training loss is not necessary (and in our case is not even well-defined).
> >
> > I understand that all the optimization and learning is conditioned on the training set,  however the bounds are independent of the learning algorithm. For the sake of clarity let’s just take an $mathcal{I}$ which is the training set (model is trained and weights are updated using it) and another $\mathcal{I}'$ as the validation set. Now lets only talk about the validation set (it could be same as the training set or different). Let $A$ be some algorithm which takes the training set and a random string (which represents randomness in NN) as input and gives the trained model $A(\mathcal{I}, r)$. I can write the validation loss as $L(A(\mathcal{l},r), \mathcal{I}’)$, i.e., compute loss on set $\mathcal{I}’$ using model $A(\mathcal{l},r)$. Now how can I use this validation loss as proxy for the generalization loss?
> >
> > Basically I am concerned about a specific scenario. Since generally training set is assumed to be sampled i.i.d from data distribution. So with some significant probability you can get a “good” training set (where the loss is very small) and loss on it could be less than the proposed lower bound because the bound is on generalization loss. Similarly with some probability a “bad” training set could be picked resulting in a very high value of loss. I found this dependency missing that is why I pointed it out.
> >
> > One point I can think of is, if one assumes the training set $\mathcal{I}$ is already given and if there is a "good" $\mathcal{I}$ (which can give very small loss) then $h_{D}$ might also reflect this (as it's also estimated using training set) and therefore will balance this somehow. But it's not clear to me if that will that happen.
> >
> >
> > >
> > > Yes, our lower bound supports attention and convolution layers in the autoencoder provided that:
> > > -  Each layer is continuous.
> > > -  The codomain of the encoder, $f(\cdot)$, is $(0,1)^{l}$.
> > >
> > > We note that our method of estimating the Lipschitz constant of the decoder $g(\cdot)$, imposes further assumptions which may or may not hold for these layer types. However, the bound is valid for the constraints provided. We do not claim tightness (see Section 8), and echo the reviewer's previous comment that the paper explains why a tighter bound is not required and it is easily usable.
> >
> > When discussing different architectures and dataset combinations, the bound does not depend on this duo. I agree that it is affected by these two independently, but is that correct? It's generally seen that some architecture types fit better for certain types of datasets; however, here the bound suggests there might be an architecture that fits best for all datasets. Any comments on why this might be?

---

> > > ### Author Response · Authors · 2025-07-22
> > > **Rebuttal by Authors**
> > >
> > > Dear Reviewer caDN,
> > >
> > > Thank you for your follow-up comments. We have provided our responses below.
> > >
> > > > I understand that all the optimization and learning is conditioned on the training set, however the bounds are independent of the learning algorithm. For the sake of clarity let’s just take an $\mathcal{I}$ which is the training set (model is trained and weights are updated using it) and another $\mathcal{I}'$ as the validation set. Now lets only talk about the validation set (it could be same as the training set or different). Let $A$ be some algorithm which takes the training set and a random string (which represents randomness in NN) as input and gives the trained model $A(\mathcal{I}, r)$. I can write the validation loss as $L(A(\mathcal{l},r), \mathcal{I}’)$, i.e., compute loss on set $\mathcal{I}’$ using model $A(\mathcal{l},r)$. Now how can I use this validation loss as proxy for the generalization loss? Basically I am concerned about a specific scenario. Since generally training set is assumed to be sampled i.i.d from data distribution. So with some significant probability you can get a “good” training set (where the loss is very small) and loss on it could be less than the proposed lower bound because the bound is on generalization loss. Similarly with some probability a “bad” training set could be picked resulting in a very high value of loss. I found this dependency missing that is why I pointed it out. One point I can think of is, if one assumes the training set $\mathcal{I}$ is already given and if there is a "good" $\mathcal{I}$ (which can give very small loss) then $h_{D}$ might also reflect this (as it's also estimated using training set) and therefore will balance this somehow. But it's not clear to me if that will that happen.
> > >
> > > Suppose that we have autoencoder $\mathcal{A}$, data distribution $\mathcal{D}$, training dataset $\mathcal{I}$, and validation dataset $\mathcal{I}'$.  Here, the generalization error is:
> > > - $\mathbb{E}_{x \sim \mathcal{D}}[ \lVert x - \mathcal{A}(x) \rVert^2]$.
> > >
> > > Furthermore, the validation loss is:
> > > - $\frac{1}{|\mathcal{I}'|}\sum_{i=1}^{|\mathcal{I}'|} \lVert x_i - \mathcal{A}(x_i) \rVert^2$,
> > >
> > > where $x_i$ are the elements of $\mathcal{I}'$. Thus, the validation loss is a sample mean estimator of the generalization error.  Even so, we have only claimed to have lower-bounded the generalization error, and so the result of Equation 4 holds always and does not depend upon any sampling process.
> > >
> > > We present the validation loss only in our numerical simulations.  While the reviewer is correct that the validation loss can mis-estimate the generalization error for small datasets, we believe that our large validation dataset  (10,000 observations sampled i.i.d.) mitigates this issue and that our estimator of the generalization error is sufficient for the limited purpose of the illustrations of Section 7.
> > >
> > > > When discussing different architectures and dataset combinations, the bound does not depend on this duo. I agree that it is affected by these two independently, but is that correct?
> > >
> > > We clarify that the bound _does_ depend on both the architecture (e.g., through latent dimension $l$) and the dataset (through data dimension $d$ and data differential entropy $h_\mathcal{D}$).
> > >
> > > > It's generally seen that some architecture types fit better for certain types of datasets; however, here the bound suggests there might be an architecture that fits best for all datasets. Any comments on why this might be?
> > >
> > > We do not claim that the MSE lower bound is attainable for any autoencoder $\mathcal{A}$, only that it is not violated. For any choice of $\eta$ (as defined in Section 4 of the paper), some architectures may achieve a lower generalization error than others, but these errors will never cross our bound, $\mathcal{F}_\eta$.

---

> > > > ### Comment · Reviewer_caDN · 2025-07-24
> > > > **Rebuttal Reply**
> > > >
> > > > >
> > > > > We present the validation loss only in our numerical simulations. While the reviewer is correct that the validation loss can mis-estimate the generalization error for small datasets, we believe that our large validation dataset (10,000 observations sampled i.i.d.) mitigates this issue and that our estimator of the generalization error is sufficient for the limited purpose of the illustrations of Section 7.
> > > >
> > > > Yes I was not concerned about the equation 4, was mostly worried about it's application. Thankyou for clearing it up.
> > > >
> > > > > We do not claim that the MSE lower bound is attainable for any autoencoder $\mathcal{A}$, only that it is not violated. For any choice of $\eta$ (as defined in Section 4 of the paper), some architectures may achieve a lower generalization error than others, but these errors will never cross our bound, $\mathcal{F}_\eta$
> > > >
> > > > Got it, this makes sense.
> > > >
> > > > Thankyou again.

---

### Decision · Action_Editor_rt9D · 2025-08-25

**Recommendation:** Accept as is

**Additional Comments:**

A solid paper, for which the reviewers unanimously recommended acceptance. The paper's strength lies in its theoretical rigor and clear presentation, making it a good contribution to the machine learning theory literature.

**Audience:**

Yes

**Audience Explanation:**

The work addresses fundamental theoretical questions in understanding autoencoders through the lens of information theory. The machine learning theory community will find this contribution valuable. The reviewers unanimously agreed on audience interest.

**Claims And Evidence:**

Yes

**Claims Explanation:**

The paper presents a rigorous information-theoretic analysis deriving a lower bound on the generalization error of autoencoders with sigmoid activation functions. The derivation is sound and well-executed. The authors provide clear proofs and address the key assumptions. The paper is transparent about limitations in practical applicability. The theoretical claims are accurate and supported by convincing evidence. The empirical demonstrations appropriately illustrate the bound's behavior and potential applications.